# Diurnal cycles of cloud cover and its vertical distribution over the Tibetan Plateau revealed by satellite observations, reanalysis datasets and CMIP6 outputs

Yuxin Zhao[1], Jiming Li[1]*, Lijie Zhang[1], Cong Deng[1], Yarong Li[1], Bida Jian[1], Jianping Huang[1]

[1]Key Laboratory for Semi-Arid Climate Change of the Ministry of Education, College of Atmospheric Sciences, Lanzhou University, Lanzhou, China

*Correspondence to*: Jiming Li (lijiming@lzu.edu.cn)

**Abstract.** Diurnal variations of cloud cover and cloud vertical distribution are of great importance to earth-atmosphere system radiative budgets and climate change. However, thus far, these topics have received insufficient attention, especially on the Tibetan Plateau (TP). This study focuses on the diurnal variations of total cloud cover, cloud vertical distribution, and cirrus clouds and their relationship to meteorological factors over the TP based on active and passive satellite observations, reanalysis data, and CMIP6 outputs. Our results are consistent with previous studies but provide new insights. The results show that total cloud cover peaks at 06:00-09:00 UTC, especially over the eastern TP, but the spatial and temporal distributions of clouds from different datasets are inconsistent. To some extent, it could be attributed to subvisible clouds missed by passive satellites and models. Compared with satellite observations, the amplitudes of the diurnal variations of total cloud cover obtained by the reanalysis and CMIP6 models are obviously smaller. CATS can capture varying pattern of the vertical distribution of clouds and corresponding height of peak cloud cover at middle and high atmosphere levels, although it underestimates the cloud cover of low-level clouds especially over the southern TP. Compared with CATS, ERA5 cannot capture the complete diurnal variations of vertical distribution of clouds and the MERRA-2 has a poorer performance. We further find that cirrus clouds, which are widespread over the TP, show significant diurnal variations with averaged peak cloud cover over 0.35 at 15:00 UTC. Unlike in the tropic, where thin cirrus (0.03<optical depth<0.3) dominate, opaque cirrus clouds (0.3<optical depth<3) are the dominant cirrus clouds over the TP. The seasonal and regional averaged cloud cover of opaque cirrus reaches a daily maximum of 0.18 at 11:00 UTC, and its diurnal cycle is strong positive correlation with that of 250 hPa relative humidity and 250 hPa vertical velocity. Although subvisible clouds (optical depth<0.03), which have a potential impact on the radiation budget, are the fewest among cirrus clouds over the TP, the seasonal and regional averaged peak cloud cover can reach 0.09 at 22:00 UTC, and their diurnal cycle is correlation with that of the 250 hPa relative humidity, 2-m temperature and 250 hPa vertical velocity. Our results will be helpful to improve the simulation and retrieval of total cloud cover and cloud vertical distribution, and further provide an observational constraint for simulations of diurnal cycle of surface radiation budget and precipitation over TP region.

## 1 Introduction

The Tibetan Plateau (TP), a heat source for the East Asian monsoon, has received worldwide attention due to its prominent altitude and special topography (Wu et al., 2017). Over the TP, surface heating causes a low-pressure centre that can attract warm-moist air convergence from the ocean and then promote convective activity (Wu et al., 2012). The abundant water storage in the atmosphere over the TP and its surrounding regions can be explained by this convective system, and the TP is thus called the "Asian water tower" (Xu et al., 2008). In recent decades, the TP has experienced significant climate warming (Liu and Chen, 2000; Yao et al., 2012), and it will continue in the future (Duan and Wu, 2006; Wang et al., 2008). The rapid warming over the TP has caused dramatic changes in the cryosphere, such as glacier retreat, snow cover reduction, permafrost degradation and expansion of glacier-fed lake areas (Cheng and Wu, 2007; Yao et al., 2007; Rangwala et al., 2010; Zhang et al., 2019). Although some studies have found that rapid warming is possibly linked to increasing surface water vapour or anthropogenic greenhouse gas emissions over the TP (Rangwala et al., 2009; Zhou and Zhang, 2021), growing evidence has verified that variations in cloud properties are also very important in determining the surface energy balance and water cycle of the TP region (Yang et al., 2012; Yan et al., 2016; Pan et al., 2017).

Indeed, cloud cover is the first-order variable impacting downwelling radiation at the surface (Naud et al., 2015), and its long-term changes over the TP and consequential influences have been explored based on satellite observations and reanalysis data (You et al., 2014; Kukulies et al., 2019). For example, based on weather observations at stations across the TP during 1961–2003, Duan and Wu (2006) found a dramatic increase in the low-level cloud amount, which ultimately led to strong nocturnal surface warming. Ma et al. (2021) also pointed out that high cloud cover is the most important influence factor on summer precipitation over the TP, based on the Clouds and Earth's Radiant Energy System (CERES) Edition 4 dataset during 2001–2009. In addition to the long-term change, however, existing studies indicate that clouds over the TP also exhibit obvious diurnal cycles. The observations from ground-based cloud radar at the Motuo National Climate Observatory over the southeastern TP show that the occurrence frequency of clouds is larger (maximum value 73%) from evening to midnight (i.e., 13:00-18:00 UTC) and reaches a minimum value (53%) in the morning (04:00 UTC) (Zhou et al., 2021). The diurnal cycle of clouds strongly affects their efficiency in regulating the radiation budget (Yin and Porporato, 2020), and it is also closely related to the diurnal cycle of precipitation (Nesbitt et al., 2008; Zhao et al., 2017). Neglecting the importance of the diurnal cycle of clouds will result in the inaccurate representation of clouds in models and exacerbate inconsistencies between observations and model simulations (Tian et al., 2004). For example, the evaluation of version 2 of the Community Climate System Model (CCSM2) shows that the simulated diurnal variations are still smaller than the observed values even if the model is driven by observational data, and the diurnal cycle of precipitation from simulations is too weak over the oceans (Dai and Trenberth, 2004). Yin and Porporato (2017) found that most General Circulation Models (GCMs) lack cloud peaks around the afternoon and thus lead to the overestimation of daily mean top-of-atmosphere (TOA) irradiance compared with ECMWF's first atmospheric reanalysis of the 20th Century (ERA-20C). Moreover, the inconsistent amplitudes and phases of the diurnal cycle of the clouds between models results in a large intermodel difference

in irradiance, which reaches a maximum of 1.8 Wm$^{-2}$ over land and 2.1 Wm$^{-2}$ over the ocean. To date, some observational studies have focused on diurnal variations of cloud cover over the TP based on geostationary satellite data and ground-based observations (Liu et al., 2015; Shang et al., 2018), and some typical features of the diurnal cloud cycle have been described, e.g., amplitude and phase (Song et al., 2017). However, satellites with passive remote sensing instruments (e.g., MODIS) generally fail to detect optically thin clouds with small optical depths (<0.3) (Minnis et al., 2008), which are found in approximately 50% of global observations (Sun et al., 2011b). These thin cirrus clouds frequently occur near the tropopause of stronger convective regions (e.g., the tropics or the Tibetan Plateau) and have great impacts on the cloud or surface properties retrieved by satellite, Earth–atmosphere system radiation energy budgets, and exchanges between the troposphere and stratosphere (Sassen et al., 2009; Sun et al., 2011a; Sun et al., 2011b; Zou et al., 2020). In addition, the thin cirrus clouds that are undetected by passive sensors also possibly contribute to part of the total cloud cover inconsistency between passive and active satellites (Stanfield et al., 2015). It is therefore of fundamental importance to study these optically thin clouds over the TP in detail, especially the relationship between their diurnal cycle and meteorological factors, to fill in related knowledge gaps in the TP region.

In addition, another important but less concerning issue in the TP region is the cloud vertical distribution, especially its diurnal cycle. The vertical profile of the cloud cover may affect the atmospheric circulation by altering the vertical gradients of the radiative heating/cooling rate and the subsequent atmospheric temperature and is also closely linked to the efficiency of precipitation production (Posselt et al., 2008; Binder et al., 2020). To date, however, too few studies have focused on the diurnal cycle of cloud vertical distribution over the TP because passive sensors onboard geostationary satellites cannot resolve the vertical structure of cloud systems. The Cloud-Aerosol Lidar and Infrared Pathfinder Satellite Observation (CALIPSO) and CloudSat satellites have a vertically resolved ability to document the cloud vertical structure at a global scale (Sassen et al., 2009; Oreopoulos et al., 2017). Although CALIPSO/CloudSat provide considerable valuable information on cloud vertical distribution, especially over areas sensitive to climate change without ground-based observations (Yan et al., 2016; Wang et al., 2021), only instantaneous cloud vertical distributions at two overpass times are possible. As a result, this study attempts to use the measurements from the Cloud-Aerosol Transport System (CATS) (Mcgill et al., 2015) onboard the International Space Station (ISS), to analyse the diurnal cycle of total cloud cover, vertical distribution and optically thin cirrus clouds over the TP. As a new space-based lidar, York et al. (2016) have indicated that CATS has similar advantages as the Cloud-Aerosol Lidar with Orthogonal Polarization (CALIOP) onboard CALIPSO for detecting optically thin clouds and cloud vertical distribution. CATS is the only space-based lidar that contains active vertical measurements with a variable local time of overpass between 51°S and 51°N and shows sufficient credibility compared to ground-based active instruments and passive and active spaceborne sensors (Noel et al., 2018). This allows us to analyse the diurnal cycle of cloud vertical distribution at regional or near-global scales based on CATS (Dauhut et al., 2020; Wang et al., 2022). In addition to CATS, this study employs cloud cover from other datasets, such as passive satellite (Himawari-8, ISCCP), reanalysis datasets (ERA5, MERRA-2) and climate model (CMIP6) outputs, to perform related comparisons. The paper is organized as follows. The data and methods used in this study are described in Section 2. Section 3 includes the comparison of diurnal variations

of total cloud cover and cloud vertical structure between different datasets. The correlation between the diurnal cycle of cirrus and meteorological factors is further discussed. Finally, the conclusions and discussion are presented in Section 4.

## 2 Data and methods

Due to sparse ground-based measurements over the TP, this investigation mainly uses multiple satellite products, reanalysis datasets and outputs of CMIP6 models to analyse the diurnal cycle of the cloud cover over the TP region (26–40° N and 73°–105° E). In addition, the TP is simply divided into four subregions by the latitude and longitude lines of 33° N and 89° E, respectively, and the boundary of the TP (Ren and Pan, 2019).

### 2.1 Cloud-Aerosol Transport System (CATS)

CATS lidar onboard the ISS is a multiwavelength elastic backscatter lidar that can measure backscattered energy profiles with nearly a three-day repeating cycle over the same locations but at different local times between 51°S and 51°N (Mcgill et al., 2015). Although CATS cannot monitor the evolution of one cloud system, its special sample mode makes it the only space-based lidar by far that provides the seasonal-averaged diurnal cycle of clouds and their aerosol properties, especially their vertical profiles, at a given location by aggregating observations at different local times of day during various 110 seasons (Noel et al., 2018). CATS employs a similar atmospheric layer-detection algorithm as CALIOP to identify the cloud/aerosol layer and retrieve layer properties (i.e., layer height and thickness, optical depth et al.) (Yorks et al., 2016), but unlike CALIOP, CATS uses the attenuated backscatter at 1064 nm instead of 532 nm because the signals at 532 nm are unavailable due to technical issues (Yorks et al., 2016). In addition, the signal-to-noise ratio at 1064 nm is higher at nighttime (Pauly et al., 2019), and the absorbing aerosol layer is more fully captured by 1064 nm (Rajapakshe et al., 2017; 115 Yorks et al., 2021). Recently, some studies have confirmed the good performance of CATS in retrieving cloud/aerosol property profiles, especially the diurnal variations, for scientific investigations (Noel et al., 2018; Yu et al., 2021).

      In this investigation, we use the related cloud layer parameters from Version 3-01 of CATS level 2 operational (L2O), 5 km layer product (L2O_D/N-M7.2-V3-01_5kmLay) during the entire period of CATS operation (March 2015–October 2017), including the 'Feature Type Score', 'Layer Base Altitude', 'Layer Top Altitude', 'Percent Opacity', 'Layer Top 120 Temperature' and 'Feature Optical Depth'. Here, we use cloudy profiles with a 'Feature Type Score' of 5 or greater to reduce the uncertainty of the layer-detection algorithm (Yorks et al., 2016; Noel et al., 2018). In addition, for a given grid (e.g., 2° ×2°), the total cloud cover based on CATS is calculated by dividing the number of cloudy profiles by the total number of profiles in each grid (Li et al., 2011; Noel et al., 2018). The cloud cover at a given height bin has a similar definition as the total cloud cover. But it is worth noting that we also use the parameter "Percent_Opacity_Fore_FOV" in the 125 CATS layer product to check the opacity of each profile. If it is not opaque, that profile contributes to the number of profiles at all altitude levels. If that profile is opaque, we don't count that profile in the total number of profiles for those altitude

levels below the base of the lowest cloud layer detected in that profile. For CATS, a profile is considered opaque if no surface return is detected in all level 1 (L1B) 350 m profiles that make up that L2O 5 km profile.

## 2.2 International Satellite Cloud Climatology Project (ISCCP)

The ISCCP dataset (Rossow and Schiffer, 1999), which is obtained from both geostationary and polar-orbiting satellite imaging radiometers with common visible and infrared channels, has been widely used to study the diurnal, seasonal and interannual variations in cloud properties (Naud and Chen, 2010; Norris and Evan, 2015; Rossow et al., 2021). Compared with the previous version of the ISCCP dataset (e.g., ISCCP-D), the newly released ISCCP-H product has many improvements (Young et al., 2018), such as a finer sampling resolution, higher data quality and expanded record period. To

date, the ISCCP-H product can provide the global total cloud cover with a $1° \times 1°$ spatial resolution every 3 hours from July 1983 through June 2017, and it has also been used in some recent studies to analyse the diurnal cycle and long-term variation in regional cloud cover (Lei et al., 2020; Zhang et al., 2020b), in model evaluation (Tselioudis et al., 2021), and in comparison with other satellite cloud products (Karlsson and Devasthale, 2018; Tzallas et al., 2019). In this investigation, we use the "cloud area fraction" parameter from the 3-hourly monthly averaged 'ISCCP Basic HGH' dataset from March 2015

to June 2017.

## 2.3 Himawari-8 geostationary satellite

The Himawari-8 geostationary satellite was launched by the Japan Meteorological Agency on 7 October 2014. The Advanced Himawari Imager (AHI), which is carried by Himawari-8, has 16 bands, including 3 visible bands, 3 near-infrared bands, and 10 other infrared channels. Based on the radiance information of these channels, the AHI can provide good-

quality cloud and aerosol products with a spatial resolution from 0.5 to 2 km and a temporal resolution from 2.5 to 10 min (Letu et al., 2018; Letu et al., 2020). Here, the "Cloud Mask Confidence Level Flag" parameter in the Level-2 (L2) operational cloud property products from January 2016 to October 2017 is used in the following analysis. The "Cloud Mask Confidence Level Flag" provided by the Himawari-8 classifies each 0.05° grid into the following four categories: clear, probably clear, probably cloudy, and cloudy. Similar to previous studies (Shang et al., 2018; Lei et al., 2020), only the

'cloudy' pixels are identified as clouds, while the others are classified as clear sky in this study. Finally, the total cloud cover given at each 0.05° grid is defined as the ratio of the cloudy sample size to the total sample size every 3 hours. In addition, until now, only cloud cover from AHI during daytime is available, thus we merely consider the period in which there are complete data over the TP, which is during 00:00 UTC to 10:00 UTC

## 2.4 Reanalysis datasets

The ERA5 reanalysis dataset from the European Centre for Medium-Range Weather Forecast (ECMWF) contains abundant variables at the surface and on single levels and pressure levels by using 4D-Var data assimilation and model forecasts of the ECMWF Integrated Forecast Systems (IFS) (Urraca et al., 2018; Hersbach et al., 2020). Cloud characteristics

from reanalysis data largely depend on atmospheric numerical models and data assimilation schemes. The physical parameterizations in ERA5 that provide the cloud properties in a grid cell are based on the advanced version of the scheme by Tiedtke (1993). In this study, the hourly total cloud cover for a single level and fraction of cloud cover at 37 pressure levels from ERA5 at a 0.25° × 0.25° resolution are used to compare the diurnal cycle of the cloud cover and its vertical distribution with other datasets. In addition, the hourly 2-m temperature, 10-m wind speed, vertically integrated divergence of moisture flux on single levels and hourly vertical velocity and relative humidity at 250 hPa on the pressure levels are used to discuss the relation between the cloud cover and meteorological parameters.

In addition to the ERA5, the cloud cover from version 2 of the Modern-Era Retrospective Analysis for Research and Applications (MERRA-2) (Rienecker et al., 2011), which has a gridded resolution of 0.5° × 0.625°, is also used. MERRA-2 is the latest atmospheric reanalysis produced by NASA's Global Modeling and Assimilation Office (GMAO). The Goddard Earth Observing System (GEOS) atmospheric model (Rienecker, 2008; Molod et al., 2015) and NCEP's grid point statistical interpolation (GSI) analysis scheme (Wu et al., 2002; Kleist et al., 2009) are the key components of version 5.12.4 of the GEOS atmospheric data assimilation system that produce MERRA-2. Specifically, this study uses the hourly total cloud area fraction from the MERRA-2 'tavg1_2d_rad_Nx' product, which is a time-averaged 3-dimensional dataset. The cloud cover for radiation at 42 pressure levels is based on the MERRA-2 'tavg3_3d_rad_Np' product, which is a 4-dimensional 3-hourly time-averaged dataset.

## 2.5 CMIP6 models

Here, we also use the 3-hourly cloud area fraction from 17 CMIP6 models with AMIP simulations that utilize observed sea surface temperatures and sea ice concentrations (Eyring et al., 2016). Assessment of the CMIP6's performance with respect to clouds is of wide concern and has achieved variable results (Cherian and Quaas, 2020; Vignesh et al., 2020). Because the temporal coverage of the historical CMIP6 outputs cannot cover the same detection period as CATS, we use the future climate simulations from March/2015 to October/2017 The scenario from Shared Socioeconomic Pathway (SSP), SSP5-8.5, is selected as it is closest to actual emissions. SSP5-8.5 is an upgrade of the RCP8.5 pathway (RCP is short for representative concentration pathway) and SSP5 assumes an energy intensive, fossil-based economy (O'Neill et al., 2016). In addition, it is worth noting that CMIP6 outputs with lidar simulator (e.g., CATS simulator) are still unavailable for the 3-hourly resolution, thus this study uses the 3-hourly cloud area fraction from CMIP6 models without lidar simulator. Due to the discrepancies in the definitions and determination algorithms of cloud cover, direct comparisons between the diurnal cycle of total cloud cover in models and satellite observations possibly results in some uncertainties (Engström et al., 2015). In the subsequent analysis, all model outputs, reanalysis, ISCCP and Himawari-8 are uniformly linearly interpolated to the 2° × 2° grid after Fig. 1 to keep consistency with CATS observation.

Table 1 lists the details of the satellite products, reanalysis datasets and model outputs in this study, including their spatial-temporal resolutions and temporal coverage.

 **3 Results**

**3.1 Comparison of the diurnal variation in total cloud cover from different datasets**

The diurnal cycle of cloud properties over the TP shows unique characteristics due to its special topography and large-scale circulation background (Wang et al., 2020). Fig. 1 shows the spatial distribution of total cloud cover over the TP for each 3-hour mean using active and passive satellite datasets, reanalysis data and climate model outputs. The statistical results from all datasets in Fig. 1 are aggregated over the entire time period of CATS observation except Himawari-8, of which the dataset over a shorter period than that of CATS is used in our study. Similar to previous studies (Shang et al., 2018; Lei et al., 2020), significant diurnal variations of total cloud cover over the TP are found in almost all datasets. The peak time and amplitude of total cloud cover both exhibit obvious differences among the different datasets due to difference in the sensitivities of detectors, cloud detecting algorithms or cloud parameterizations. For CATS, clouds have a maximum coverage in the early afternoon (e.g., 06:00 UTC), and cloud cover can reach a daily maximum of 0.8 over the central and eastern TP. However, a daily minimum total cloud cover (mean value about 0.4) over the southwestern TP is found at 00:00 UTC. The diurnal variation of total cloud cover from the ISCCP is similar to that of CATS but ISCCP usually exhibits an obvious smaller total cloud cover during night (e.g., from 12:00 UTC to 00:00 UTC) than that from CATS. Compared with the ISCCP, the higher total cloud cover over the eastern TP (especially at night) detected by CATS might be related to the subvisible or optically thin cirrus clouds (also see Fig. S2b and S2c), which are more frequent during the night and usually misclassified as clear sky by passive sensors or satellites such as those of the ISCCP, MODIS, and MISR because their minimum detectable cloud optical thickness is approximately 0.1 to 0.4 (Marchand et al., 2010). Indeed, by comparing the total cloud cover from the Aqua/MODIS and CALIOP, Holz et al. (2008) found that the cloud detection results from the MODIS and CALIOP agreed more than 87% of the time, and their discrepancies were largely associated with the optically thin clouds that were undetected by MODIS but that were readily observed by CALIOP. From a global mean perspective, the optically thin clouds resulting in the total cloud cover from CALIPSO–CloudSat are approximately 10% higher than those from the CERES–MODIS (Stanfield et al., 2015). Sun et al. (2011b) also pointed out that if these optically thin clouds were completely mistaken for clear sky, approximately 15 Wm$^{-2}$ of the heating effect would be missed. The total cloud cover detected by the Himawari-8 satellite is nearly half that of CATS, except at 09:00 UTC. This difference may be partly related to the detection limitation of the Himawari-8 and its strict cloud identification algorithm (Imai and Yoshida, 2016). For the higher spatial resolution Himawari-8 and ERA5 data, the total cloud cover shows a higher value on south-facing slopes, likely caused by the small cumulus growth (Shang et al., 2018). However, this phenomenon is not obvious in other datasets, probably because of the difference in resolution. For reanalysis, the daily amplitudes of the total cloud cover of the ERA5 and MERRA-2 are both relative smaller than those of satellite datasets, and the total cloud covers of these two reanalysis datasets are almost lower than the results of CATS all the day. However, ERA5 shows a higher cloud cover over the Linzhi region of Tibet from 18:00 UTC to 00:00 UTC than those from CATS, ISCCP and MERRA-2 datasets. It could be the ERA5 overestimates low-level clouds, which will be explained in more detail in Section 3.2, which discusses the vertical

distribution of clouds. In addition, we find that the total cloud cover between two reanalysis datasets also exhibits considerable difference regardless of magnitude or peak time of total cloud cover. Generally speaking, total cloud cover from

MERRA-2 is lowest among these datasets except Himawari-8 satellite, which has a smaller total cloud cover over the northern TP before 06:00 UTC than that of MERRA-2, meantime, the maximum of total cloud cover from MERRA-2 does not exceed 0.6 over all subregions. From a global perspective, Li et al. (2017) indicated that the MERRA-2 underestimates total cloud cover nearly everywhere compared with the CERES–MODIS dataset. The diurnal cycle of total cloud cover of the multimodel in CMIP6 (hereafter MEM) shows some similarities with that of the ERA5, for example, MEM also

simulates the high total cloud cover over the Linzhi region of Tibet during night, and produces similar daily amplitude of the total cloud cover. However, MEM gives larger total cloud cover than those from reanalysis and Himawari-8. At a given time, MEM also exhibits similar spatial distribution of total cloud cover with that of CATS, but underestimation from MEM is still very obvious over the most part of TP region. Besides reanalysis, the results show that the amplitude of the climate model simulations also differs significantly from the satellite results over the TP. This problem exists on a global scale and induces

overestimation of radiation in most climate models (Yin and Porporato, 2017). Of course, the comparison of different datasets may be biased due to discrepancies in the definitions and determination algorithms of cloud cover, and it may also be limited by the fact that the sampling location of each dataset cannot totally coincide.

Due to the different distributions of temperature, moisture, etc., Fig. 1 clearly shows that the distributions of total cloud cover also exhibit obvious spatial variability at different local times. Here, the TP is simply divided into four subregions

along the latitude and longitude lines of 33° N and 89° E (shown in total cloud cover from CATS at 00:00 UTC in Fig. 1), respectively, and diurnal variations of regional averaged total cloud cover over these subregions are provided in Fig. 2. It is worth noting that this is only a simple and rough zoning method without a detailed matching of dynamic and circulation structures, which could have an impact on the results, but it can also reflect the differences between the monsoon-controlled area and the non-monsoon area on the TP to some extent (Yao et al., 2013). Over the northwestern TP (Fig. 2a), the range of

the diurnal total cloud cover detected by CATS is approximately 0.54-0.79, and the peak time is around 06:00 UTC. Compared with CATS, the ISCCP exhibits a higher total cloud cover during the daytime (e.g., from 03:00-12:00 UTC) and a later peak time at approximately 09:00 UTC (maximum value around 0.85). After 12:00 UTC, the total cloud cover in the ISCCP is obviously less than that of CATS, and its minimum value and daily range are approximately 0.48 and 0.37, respectively. From a regional mean perspective, ERA5 and MERRA-2 have comparable daily amplitude of the total cloud

cover with that of MEM, and its values are 0.17, 0.16 and 0.13 for ERA5, MERRA-2 and MEM, respectively. Although reanalysis datasets have similar peak (e.g., around 10:00 UTC) and valley (e.g., 03:00 UTC) times with that of MEM, the diurnal variations of total cloud cover from MERRA-2 and MEM are almost synchronous, and their peak (or valley) values are approximately 0.54 (or 0.38) and 0.64 (or 0.51), respectively. In our study, the Himawari-8, which can detect the total cloud cover of the TP only between 00:00 UTC and 10:00 UTC, shows a comparable significant amplitude (about 0.37) of

diurnal cycle of total cloud cover with that of ISCCP and is nearly one and a half times as large as that of CATS (Fig. 2a). Over the northeastern TP (Fig. 2b), the diurnal cycle of total cloud cover from different datasets exhibits smaller amplitude

compared with those over the northwestern TP, and the amplitudes from the reanalysis and MEM are both less than 0.1 and only one-third of those of satellite datasets, especially for ERA5 (amplitude around 0.07). For the southwestern TP, total cloud covers in all the datasets over this subregion are always the lowest in all subregions (Fig. 2c), possibly limited by the moisture flux and the high terrain of the Himalayas. However, the diurnal amplitudes of total cloud cover from all datasets are comparable with those over the northwestern TP. Over this subregion, lowest total cloud cover is produced by MERRA-2 instead of Himawari-8, and MEM even simulates more cloud cover than CATS at some hours of night. Similar results are also found over the southeastern TP (Fig. 2d). On average, diurnal amplitude of total cloud cover in most datasets is smaller over the eastern TP than that over the western TP, and ISCCP and MEM produce maximum and minimum diurnal amplitudes, respectively.

To find out in which regions the diurnal cycle of which datasets are well correlated with CATS, Figure 3 further shows the spatial distribution of correlation coefficients of diurnal cycle for total cloud cover between CATS and other datasets in a $2 \times 2°$ grid box. As shown in the Fig. 3, ISCCP exhibits best correlation with CATS, and the correlation coefficient (at 90% confidence level) is even greater than 0.5 over the most areas (Fig. 3a), especially over the central part of TP. The diurnal cycle of total cloud cover from the Himawari-8 obviously positive correlates with that of CATS over the most part of TP, but the correlation is almost insignificant over TP region (Fig. 3b). It may be caused partly by the limited observation hours from Himawari-8. Here, it is worth noting that because the cloud cover calculation of CATS needs to ensure that there are enough profiles in each grid, it is difficult to split more sample points by months or seasons for correlation analysis. Therefore, the correlation analysis here can only be used as a reference to some extent. Similar with ISCCP, ERA5 also shows significant positive correlation with diurnal cycle of total cloud cover of CATS over the central and western parts of TP (see Fig. 3c), but we also find the ERA5 is the only dataset that exhibits opposite diurnal variation with CATS over the eastern part of TP, and correlation coefficient (at 90% confidence level) even reaches -0.9. As stated in Fig. 2, MERRA-2 and MEM show almost synchronous diurnal variations of total cloud cover, resulting in the correlations coefficients of diurnal cycle from them with CATS are very similar, that is, there is a significant positive correlation coefficient over the northern part of TP (Fig. 3d and 3e). Although Fig. 3 indicates that ISCCP exhibits closer diurnal cycle of total cloud cover with that of CATS over most part of TP, the averaged spatial consistency of total cloud cover at all times between ISCCP and CATS is lowest compared with those from ERA5, Himawari-8, MERRA-2 and MEM (see Fig. A1 in the appendix). In summary, above statistical results show that total cloud cover from multiple sources exhibits considerable regional differences in the phase and magnitude of the diurnal cycle.

As stated in Section 2, the above total cloud cover differences in satellite datasets partly refer to the detection limitations of different sensors. By comparing the total cloud cover estimated from the CloudSat-CALIPSO with the ISCCP, Naud and Chen (2010) pointed out that the total cloud cover from the ISCCP is underestimated by approximately 18% over the TP during night, in part due to the misdetection of low-level clouds at night, which can partly explain the difference between the ISCCP and CATS at night in our study. However, optically thin clouds are also important contributors to large differences in cloud cover between the ISCCP and CATS, especially during summer, when high-level, optically thin clouds

occur more often than in the other seasons (Naud and Chen, 2010). In addition, Naud and Chen (2010) also pointed out that cloud property retrieval from the ISCCP is more consistent with those from the CloudSat-CALIPSO over high-elevation regions of the TP where multi-layered cloud systems are infrequent. In fact, when the optically thin clouds overlap with other cloud types, the passive satellite will bias the cloud top properties of the underlying clouds (e.g., cloud top temperature

or height), but the total cloud cover is almost unaffected. However, if only optically thin clouds are present, the ISCCP easily misclassifies them as clear sky. Thus, multi-layered cloud systems are not the main contributors to the total cloud cover differences between the ISCCP and CATS. Besides, ISCCP sometimes also overestimates the total cloud cover during daytime compared with CATS. By comparing the spatio-temporal matched total cloud cover from ISCCP, CALIPSO alone and the combined product from CALIPSO and CloudSat (that is, 2B-GEOPROF-lidar) during daytime, we find that ISCCP

still overestimates the total cloud cover over TP compared with those of other space-based lidar and radar (figure not shown). Similar, Boudala and Milbrandt (2021) also found that ISCCP has larger cloud cover than that of CALIPSO over mid-latitudes (e.g., the European continent). Tzallas et al. (2019) noted that the larger cloud cover of ISCCP in the European continent is link to the relatively large viewing zenith angle (VZA) of ISCCP.  Knapp et al. (2021) also suggested that there is a VZA dependence in the cloud cover of ISCCP. Previous studies have shown that spurious detection or missed detection

of clouds is the largest source of systematic errors in ISCCP results. For ISCCP, it is difficult to distinguish between aerosols and thin cirrus clouds, which may lead to spurious cloud detections and thus to an overestimation of clouds (Rossow and Schiffer, 1999). In addition, our study also find that larger differences exist between the ISCCP (or the ERA5) and the Himawari-8, especially at 03:00 UTC. Using similar datasets, Lei et al. (2020) showed that the ERA5 overestimates approximately 10% and the ISCCP overestimates 20% of the total cloud cover over the TP compared to the Himawari-8.

However, our results indicate that the ERA5 and ISCCP have more closer cloud cover with that from CATS compared with that of Himawari-8. It means that Himawari-8 should underestimate the total cloud cover than ERA5 and ISCCP before 09:00 UTC at least.

## 3.2 Comparison of cloud vertical distribution from different datasets

The vertical structure of clouds is closely related to precipitation and cloud radiative effects, which has attracted

widespread attention (Wang et al., 2000; Yan et al., 2016; Li et al., 2018). Moreover, the cloud vertical structure is an important factor in studying how climate change influences cloud feedback (Wang et al., 2000; Bodas-Salcedo, 2018). Until recently, information about cloud vertical structure was usually extracted from surface observations, such as radiosonde data, which can provide four of five decades of records for climate research (Wang and Rossow, 1995). Since the launch of CALIPSO and CloudSat, active satellite data have been widely used in the study of global cloud vertical structures

(Oreopoulos et al., 2017). However, the limited observation times each day from the CALIPSO/CloudSat result in an unclear diurnal cycle of cloud vertical distribution over the TP. In a recent study, Noel et al. (2018) first used CATS to analyse the diurnal cycle of cloud profiles over land and oceans between 51°S and 51°N and found similar vertical distributions of cloud cover between CATS and CALIPSO at a near-global scale. In this section, we compare the cloud vertical distribution of

different subregions using CATS, ERA5 and MERRA-2. As a reference, the spatio-temporal matched cloud vertical profiles

from CALIPSO alone and the combined product from CALIPSO and CloudSat (that is, 2B-GEOPROF-lidar, marked as CALIPSO & CloudSat in Fig. 4) over the entire observation period of CATS are also used. Here, the calculation of cloud cover at a given height bin is same as that of CATS, that is, removing the profiles that are fully attenuated below opaque layers from the total number of profiles (see the section 2.1)

Fig. 4 provides the cloud vertical profiles at the hour closest to the CloudSat and CALIPSO daytime overpass time, and

330 the results during different seasons at this time are given in Fig. S1. In this study, all the altitudes are above the mean sea level. In addition, we also add the topmost and bottommost surface height altitudes of each region in the Fig. 4 based on the DEM elevation information in CATS L2O Layer products. From the averaged cloud vertical distribution over the whole TP region (Fig. 4a), CATS, CALIPSO and CloudSat & CALIPSO exhibit similar peak heights (approximately 7-8 km) of cloud cover, whereas ERA5 and MERRA-2 slightly overestimates the peak height of cloud cover (around 9 km). Similar to Noel et

al. (2018), we also find that the cloud vertical profiles from CALIPSO agree well with those from CATS, especially below the peak height over the northwestern and southwestern parts of the TP (Fig. 4b and Fig. 4d). The small negative difference between CATS and CALIPSO possibly comes from the spatio-temporal matching process, mostly. Such agreement between CATS and CALIPSO is understandable because CATS employs an atmospheric layer detection algorithm similar to that of CALIOP (Yorks et al., 2016). This means that they also have similar detection limitations; for example, they cannot

penetrate optically thick clouds to detect the underlying clouds and thus underestimate cloud cover at low atmosphere levels. The underestimation of low-level clouds by CATS and CALIPSO due to optical extinction from higher clouds can be slightly improved via removing those profiles that are fully attenuated below opaque layers from the total number of profiles. The 2B-GEOPROF-lidar combines the advantages of CALIPSO and CloudSat in detecting both optically thin and thick cloud systems and thus can provide a relatively accurate cloud vertical distribution compared to other datasets. Compared

with 2B-GEOPROF-lidar, however, we find that CATS and CALIPSO datasets still obviously underestimate the cloud cover at middle and low atmosphere levels even if we remove those profiles that are fully attenuated below opaque layers from the total number of profiles, and the bias of cloud cover even reaches 0.2 and 0.15 at 8 km and 4 km over the southeastern TP (Fig. 4e), respectively. In particular, the bias between CATS (or CALIPSO) and the 2B-GEOPROF-lidar product is more obvious during the spring and summer seasons (Fig. S1). At the middle level of the atmosphere, cloud cover differences

between CATS (or CALIPSO) and 2B-GEOPROF-lidar may result from altocumulus, altostratus or deep convective clouds. However, at the low level of the atmosphere, their cloud cover difference mainly comes from the undetected cumulus or stratus clouds in CATS due to lidar signal attenuation, especially over the southeastern part of the TP (Fig. 4e), where surface wind convergence and upwards motion forced by topography tend to promote cumulus clouds (Li and Zhang, 2016).

The peak cloud covers from the reanalysis datasets are obviously lower than those detected by active satellites.

Moreover, the peak heights and cloud covers are also different between the reanalysis datasets over the southern part of the TP (Fig. 4d and 4e). The ERA5 has a peak cloud cover (approximately 0.2) at 8 km over the western TP, whereas the peak height exceeds 9 km over the northeastern TP. Fig. 4a clearly shows that an important peak cloud cover exists at low

atmosphere levels (<4 km) in the ERA5 over the whole TP. This peak is particularly obvious over the southern part of the TP (Fig. 4d and 4e). Over the southwestern TP (Fig. 4d), the ERA5 obviously overestimates the cloud cover compared with the 2B-GEOPROF-lidar below 4 km, but the other datasets maintain a consistently low cloud cover, especially during the summer and autumn seasons (Fig. S1c and S1d). One possible cause might be that the very large terrain causes a very large model bias in the vertical cloud distribution of the ERA5 model, which is also found in the ERA-interim (Yin et al., 2015). Over the southeastern TP (Fig. 4e), although ERA5 and 2B-GEOPROF-lidar both exhibit larger cloud cover below 4 km compared with those from other datasets, the cloud cover from ERA5 is still obvious larger than that of 2B-GEOPROF-lidar. In contrast to the cloud cover retrieval by satellites, the ERA5 prognosticates a grid box fractional cloud cover by parameterizing cloud formation and evolution processes that consider cumulus updraughts, vertical motions, diabatic cooling, etc. (ECMWF, 2016). By comparing the vertical cloud structure of warm conveyor belts in the ERA5 and CloudSat/CALIPSO datasets, Binder et al. (2020) found that the ERA5 represents the frozen hydrometeor distribution well but underestimates the high ice and snow values in the mixed-phase clouds near the melting layer. In addition, they point out that many small and mesoscale structures observed by remote sensing instruments are not captured by the ERA5. Similar to total cloud cover, the vertical profile of cloud cover in the MERRA-2 is also obviously underestimated, but the height of the peak value is significantly overestimated over eastern part of TP (Fig. 4c and 4e). This phenomenon is also found in the tropics, where the MERRA-2 succeeds in representing high-level clouds but dramatically underestimates low- and mid-level clouds compared with CALIPSO/CloudSat (Miao et al., 2019). The poor specification or parameterization of critical relative humidity, which is the humidity threshold for cloud formation in the estimation of cloud cover in MERRA-2, is possibly responsible for the bias (Molod, 2012; Yeo et al., 2022). Finally, these biases will exacerbate the uncertainties in the heating rate profiles of MERRA-2 and ERA5.

### 3.3. Diurnal cycle of cloud vertical distribution

After realizing the intrinsic uncertainties of different datasets in characterizing the cloud vertical distribution, the diurnal cycles of cloud vertical distribution over different subregions from CATS, ERA5 and MERRA-2 are further compared in Fig. 5. Fig. 4 shows a significant underestimation of cloud cover in the ERA5 and MERRA-2 datasets compared with CATS at a given time. In fact, the underestimation is persistent throughout the day (see Fig. 5). Over the northern part of the TP, clouds are distributed in a relatively narrower height range (e.g., from 4 km to 14 km) than those over the southern part of the TP (e.g., from 4 km to 18 km), which may be linked to the deep convective clouds or cirrus clouds over the southern part. Over the northwestern TP, the cloud cover at approximately 8 km reaches its maximum value (approximately 0.3) of the day, which is sustained from 06:00 UTC to next 01:00 UTC for CATS. In the ERA5, the maximum cloud cover at approximately 8 km is approximately 0.2, and this value is sustained only from 09:00 UTC to 18:00 UTC. Considerable differences in the cloud cover between CATS and ERA5 mainly occur during nighttime. Two points need to be emphasized. First, the ERA5 captures clouds above 12 km from 12:00 UTC to 21:00 UTC, which is consistent with CATS. Second, the ERA5 also exhibits more low-level clouds below 4 km during the night. This

phenomenon is particularly obvious over the southern part of the TP. For example, low-level clouds have a maximum of 0.25 over the southeastern TP and persist from 12:00 UTC to next 03:00 UTC. Recent ground-based cloud vertical structure observations indicate that the cloud base height has an obvious peak at 1.5 km-3.5 km for the whole day, and the frequency is greater at night in the dry seasons over the southeastern TP (Zhou et al., 2021). The formation of low clouds over the TP is

thought to be favoured by low air density and strong turbulence (Xu, 2012) and associated with large-scale convergence and planetary boundary layer processes (Li and Zhang, 2016). By using CloudSat–CALIPSO datasets, Kukulies et al. (2019) also point out that the stratocumulus and cumulus clouds dominate low-level clouds over the TP, and there are more stratocumulus clouds during the monsoon season (May to September) and more cumulus clouds during the westerly season (October to April). Although we already remove those profiles that are fully attenuated below opaque layers from the total

number of profiles, this result still verifies the inability of CATS to detect the vertical structure of low clouds or optically thick clouds, but the persistent low clouds over the southwestern TP in the ERA5 may also be problematic (also see Fig. 4d). In addition to low-level clouds, the ERA5 and MERRA-2 models also cannot reproduce the distribution nor the magnitude of the diurnal cycle of cloud cover at approximately 8 km over the southern TP. Here, it is worth noting that CATS observes a high cloud cover over the southwestern TP between 11-14 km at approximately 13:00 UTC, it mainly due to the total

sample number at this hour is obvious less than those of other hours. Thus, this result is not as robust as other times. Compared with the ERA5, the MERRA-2 model has a poorer performance in reproducing the diurnal cycle of cloud cover, with larger biases in both the maximum cloud cover and its height. Here, it is worth noting that although CATS also obviously underestimates the cloud cover at almost every height compared with the 2B-GEOPROF-lidar (see Fig. 4), it can still capture the pattern of the cloud vertical distribution and the corresponding height of the peak cloud cover (see Fig. 4).

This means that a qualitative assessment of cloud cover at middle- and high atmosphere levels (e.g., peak cloud cover at 8 km) from reanalysis products with CATS observations is still feasible. In addition, the red lines at the top of Fig. 5 represent the diurnal variation in tropopause height, which is obtained from CATS level 2 operational (L2O) 5km profile products. The original data is provided by MERRA-2 reanalysis data, which is interpolated to the CATS 5 km L2O horizontal resolution (see CATS L2O Profile Products Quality Statements: Version 3.00, available online at

https://cats.gsfc.nasa.gov/media/docs/CATS_QS_L2O_Profile_3.00.pdf). Similar with total cloud cover, we gather the tropopause height information from all profiles in each subregion and calculated the hourly average over the entire observation period of CATS. Here, the tropopause height is used to find out how many clouds can penetrate the tropopause over the TP, which is a large terrain with a high altitude. The diurnal variation in the clouds overshooting the tropopause will be explored in Fig. 7 in the next section.

**3.4 Diurnal variations of cirrus and overshooting clouds**

Due to the high sensitivity of lidar signals to cirrus clouds, space-based lidar is considered an irreplaceable tool in detecting cirrus clouds and their vertical distribution at a global scale, especially in the upper troposphere and lower stratosphere (Fu et al., 2007; Virts et al., 2010). As an important cloud type, cirrus clouds play an important role in

influencing the Earth's radiation budget and accurately calculating the heating rate (Liou, 1986; Hartmann et al., 2001). A

425 recent study suggests that changing the physical properties of cirrus clouds may even counteract global warming. By seeding cirrus clouds with efficient ice nucleating particles, which may shorten their lifetime and make them more transparent, the increase in global mean surface temperature projected with $1.5 \times CO_2$ concentrations is counteracted by 70% in the CESM–CAM5 model (Gasparini et al., 2020). In addition, cirrus clouds can affect ozone concentrations in the upper troposphere and lower stratosphere (UT/LS) by acting as a potential surface for heterogeneous reactions (Borrmann et al., 1996). Similar to

430 the cirrus classification method of Sassen et al. (2009), this study defines cirrus clouds as clouds whose cloud top temperature is less than -40 °C. Based on their optical thickness (τ), cirrus clouds may be further divided into three types: subvisible cirrus (τ<0.03), thin cirrus (0.03<τ<0.3) and opaque cirrus (0.3<τ<3) clouds (Sassen and Cho, 1992). Previous studies have investigated the radiative effect of these different cirrus cloud types. For example, Fusina et al. (2007) found that the differences in heating rates between thin cirrus clouds and ice supersaturated regions can reach up to 15 K d$^{-1}$ at the

435 meteorological observatory in Lindenberg, Germany. By matching the observations of CERES, MODIS and CALIPSO satellites, Sun et al. (2011b) point out that cirrus clouds whose optical depth is less than 0.3 have a significant cooling effect on shortwave radiation by increasing the diurnal mean reflected shortwave flux by approximately 2.5 W m$^{-2}$, and clouds with an optical depth of 0.1 can have a warming effect of approximately 15 W m$^{-2}$. In addition, the subvisible cirrus clouds with significant positive radiative forcing have recently received much attention after the establishment of many effective

detection methods (Sun et al., 2014; Sun et al., 2015).

Until now, however, few studies have focused on the diurnal cycle of cirrus clouds, especially over the TP region. In the simulation of the life cycle of anvil cirrus clouds, Gasparini et al. (2019) found that adding the diurnal variations of solar radiation would affect the evolution and radiative effects of cirrus. In this section, we further use the observational advantage of CATS to discuss the diurnal cycle of cloud cover of cirrus clouds with different optical depths over the TP region (Fig. S2). In addition, the seasonal variations and diurnal cycle of regional averaged cloud cover for different cirrus types are

445 provided in Fig. 6. Opaque cirrus clouds (Fig. 6d) are found to be the main components of total cirrus clouds (Fig. 6a), and their diurnal cycles are similar. The peak cloud cover of opaque cirrus clouds occurs at 11:00 UTC (peak value is about 0.18), especially over the northeastern TP, where its value can reach 0.24 (see Fig. S2d). Over the southwestern TP, opaque cirrus occurs less frequently than over the central and northeastern parts. After 14:00 UTC, opaque cirrus clouds gradually decrease

and have a minimum value of 0.07 at 03:00 UTC (see Fig. 6d). Opaque cirrus clouds usually occur more frequently during the spring season and less frequently during autumn. In addition, their daily range even exceeds 0.2 during spring (Fig. 6d). Compared with other cirrus cloud types, the cloud cover of subvisible cirrus clouds is smallest, and its peak time is obvious later than that of opaque cirrus clouds (Fig. 6b). On average, subvisible cirrus clouds have a maximum regional averaged cloud cover (approximately 0.09) at 22:00 UTC and a minimum value of 0.02 at 04:00 UTC. Be different from that of

opaque cirrus clouds, among all seasons, the cloud cover of subvisible cirrus clouds is the largest in the summer season with the peak at 21:00 UTC. And similar peak value (approximately 0.125) occurs at 22:00 UTC in the spring season (see Fig. 6b). These statistical results above show that subvisible and opaque cirrus clouds over the TP are more frequent during nighttime

and daytime, respectively. Based on the limited observations from two overpass times of the CALIPSO and CloudSat satellites, Sassen et al. (2009) also indicate that opaque cirrus clouds are generally found during the daytime, whereas subvisible cirrus clouds are mainly found at night. For thin cirrus clouds, although their diurnal cycle is not as significant as that of the other two kinds of cirrus clouds (Fig. 6c and Fig. S2c), the maximum cloud cover at 22:00 UTC and minimum cloud cover at 03:00 UTC are still obvious. Generally, there are more cirrus clouds in spring, and the cloud cover of total cirrus clouds is the smallest in autumn. Moreover, the diurnal cycles of different cirrus cloud types cancel each other and decrease the amplitude of the diurnal cycle of cirrus clouds, especially at 09:00-23:00 UTC (Fig. 6a). On average, the cirrus cloud cover over the TP is mainly comprised of opaque cirrus clouds, followed by thin cirrus clouds, and subvisible cirrus clouds are the least common. This result is different from those of other regions (e.g., northern South America, equatorial Africa, and the western Pacific) according to Sassen et al. (2009), who found that the thin cirrus category comprises the majority of global cirrus clouds, followed by subvisible cirrus clouds, and opaque cirrus clouds are the least common. These results also reflect the regional difference in different cirrus types, which is possibly linked to several potential cirrus cloud formation mechanisms (e.g., radiative cooling in moist upper-tropospheric layers, convective blow-off, and temperature perturbation caused by convective activity such as gravity waves) (Ramaswamy and Detwiler, 1986; Heymsfield et al., 2017; Zhang et al., 2020a).

Previous studies indicate that some clouds can penetrate the tropopause into the stratosphere, especially over the TP, where tropopause folding events can reach 80% during certain winters (Chen et al., 2011), and these events are always accompanied by overshooting convective systems (Tian et al., 2020). Overshooting clouds driven by convective activity can affect the material exchange between tropospheric and stratospheric signals (Tian et al., 2011). Both water vapor and oxidation of stratospheric methane directly transported from the troposphere contribute to the increase in stratospheric water vapor. On the one hand, increasing stratospheric water vapor exacerbates the greenhouse effect (Forster and Shine, 2002). On the other hand, stratospheric water vapor can be transported to high latitudes by large-scale meridional circulation (e.g., Brewer-Dobson circulation) (Butchart, 2014). In the polar regions, the stratospheric water vapor concentration determines the critical temperature below which heterogeneous reactions on cold aerosols become important (the mechanism driving enhanced ozone depletion) and the temperature of the Arctic vortex itself, thus increasing stratospheric water vapor also enhances polar ozone consumption (Kirk-Davidoff et al., 1999; Luo et al., 2011). In particular, the impact of overshooting convection on stratospheric water vapour depends on the hour timescale (Dauhut et al., 2020). By following the methods of Dauhut et al. (2020), we perform overshooting detection on individual profiles. Because the cloud-aerosol discrimination algorithm cannot be applied to the CATS L2O layer entirely above the tropopause (Pan and Munchak, 2011), we only consider the cloud with base lower than tropopause height and top higher than tropopause height as overshooting cloud as did by Dauhut et al. (2020). Based on the CATS data, Dauhut et al. (2020) explored the diurnal cycle of tropical overshooting clouds and found that cloud cover has a first peak at 19:00 or 20:00 LT (local time) and a second peak around 00:00 or 01:00 LT. Here, to add the robustness of statistical result, we combine the all samples in subregions and provide the diurnal cycle of overshooting cloud cover over the whole TP (Fig. 7). Related regional and seasonal results, please see the

Fig. S3. Over the TP, the averaged cloud cover of overshooting cloud is higher at night and has a maximum value at 16:00 UTC (22:00 LT), and its value is about 0.013 (Fig. 7). The overshooting cloud cover over the TP is smaller than that in the tropics (Dauhut et al., 2020) with one order of magnitude. Sun et al. (2021) also found this difference in magnitude of occurrence frequency of convective overshooting between TP and tropical and subtropical areas based on TRMM. Besides the 16:00 UTC, overshooting cloud cover also has large value around 10:00 UTC (16:00 LT), 13:00 UTC (19:00 LT), 20:00 UTC (02:00 LT) and 22:00 UTC (04:00 LT). Multiple peaks in diurnal cycle are possibly caused by the regional difference of overshooting cloud. Such as, peak value at 16:00 UTC is linked with the overshooting cloud over the southern TP (Fig. S3a), especially during the summer (Fig. S3b). The peak value at 13:00 UTC is possibly related with the overshooting cloud over the southeastern and northwestern parts of TP (Fig. S3a), especially during the winter (Fig. S3b). Here, it is worth noting that the seasonal and regional results in Fig. S3 are not robust as those in the Fig. 7 due to fewer cloudy sample (see Fig. S4). However, even if Fig. 7 reveals the diurnal cycle of overshooting cloud cover over the whole TP to a certain extent, the statistical result is still noisy due to the overshooting cloud sample number only approaches 300 at 16:00 UTC and is less than 100 most of time (see Fig. S5). In addition, the difference in the tropopause altitude from different data source also possibly induces some uncertainties in our statistical result. For example, by comparing the tropopause height from MERRA-2, ERA5 and COSMIC observation data, sun et al.(2021) pointed out that the spatial distribution of tropopause height from COSMIC and ERA5 are similar, but the tropopause from MERRA-2 is a little higher than COSMIC. Overall, the differences between the tropopause height from MERRA-2 and ERA5 is within 0.6 km over the TP (Sun et al., 2021). It means that the overestimation in tropopause height from MERRA-2 may cause a little bit underestimation of overshooting cloud cover over TP. It is the one of possible reasons why the overshooting cloud cover over the TP is smaller than that in the tropics by Dauhut et al. (2020), who used ERA5 temperature and pressure profiles to compute the tropical tropopause height.

## 3.5 Meteorological factors associated with total cloud cover and cirrus clouds

The diurnal variation in cloud cover is closely related to the diurnal variations of meteorological fields, which promote or inhibit cloud formation (Feofilov and Stubenrauch, 2019; Lei et al., 2020). In this section, we further analyse the correlation of the diurnal cycle between the total cloud cover (and cirrus cover) from CATS dataset and related meteorological factors in the ERA5 dataset over the TP. All factors, including total cloud cover and meteorological factors, are standardized using z-score transformation for comparison. Z-scores measure the distance of a data point from the mean in terms of the standard deviation. This method is used for the comparison of datasets with different units and retains the shape properties of the original datasets (same skewness and kurtosis). Fig. 8 indicates that the total cloud cover, 2-m temperature and 10-m wind speed almost always peak in the afternoon (approximately 09:00 UTC) regardless of region, although their peaks do not coincide perfectly. Conversely, the vertically integrated divergence of moisture flux reaches its daily lowest value in the afternoon (approximately 09:00-12:00 UTC). Among all factors and regions, the correlation between the total cloud cover and the vertically integrated divergence of moisture flux is the strongest over the northeastern

TP (Fig. 8b), with a correlation coefficient of -0.75. The absolute values of the correlation coefficients between the meteorological factors and the total cloud cover all exceed 0.4, and all of them pass the 90% significance test. The correlation coefficients suggest that the total cloud cover is strongly correlated with the 2-m temperature, 10-m wind speed, and vertically integrated divergence of moisture flux, regardless of region. In fact, the relationship between the diurnal variations of cloud cover and meteorological factors can be explained mainly by the dynamic and thermal processes of cloud

formation and involves processes at different levels of the atmosphere (Kuang and Bretherton, 2006). For example, previous studies have indicated that strong wind near the surface facilitates the transport of moist air at low levels, whether it comes from the Indian Ocean in winter or from the surrounding convergence in summer (Yan et al., 2016). Abundant water vapour is beneficial to cloud formation, which also explains the influence of the vertically integrated divergence of moisture flux on cloud cover. In addition, solar warming of the surface powers the lifting of air masses, which can produce a buoyantly

unstable layer near the surface and promote cloud formation, especially of convective boundary layer clouds (Angevine et al., 2001). However, we know that these dynamic and thermal processes between clouds and meteorological factors are coupled, which means that meteorological factors are both linked to the formation of clouds and affected by the clouds (Betts et al., 2014). Thus, the correlation analyses above provide only limited insights into the effects of different meteorological parameters on the total cloud cover diurnal cycle, but they cannot be used to prove a robust causal relationship between them.

The diurnal variations of cloud cover and meteorological factors vary at the lower and upper tropopause and are driven by different mechanisms (Chepfer et al., 2019). Using ground-based remote sensing data, Mace et al. (2006) found that cirrus clouds are more likely to form in the ascending region of the upper troposphere during the cold season, and in summer, the formation of cirrus clouds is also always linked to detrainment from deep convection with both vertical motion and humidity anomalies. The detrainment from deep convection accompanied by small-scale condensate mass updrafts can form

cirrus clouds (Mace et al., 2006). In addition, mid-latitude weather disturbances with gentle ascending motion are associated with the formation of cirrus clouds (Heymsfield, 1977), and the generation of local convective instabilities also promotes cirrus formation (Sassen et al., 1989). Thus, cirrus formation mechanisms include the supersaturating of water vapour caused by the lifting of the air parcel (e.g., large-scale front, small-scale vertical circulations, convective clouds, and gravity waves) or by radiative cooling (Heymsfield et al., 2017). The above formation mechanisms of cirrus clouds are partly linked to

related meteorological variables (e.g., 250 hPa relative humidity, 2-m temperature, and 250 hPa vertical velocity). Thus, the relationship between the diurnal variations of regional averaged cirrus clouds and these parameters is explored in Fig. 9. All factors are standardized using z-score transformation as in Fig. 8 and the higher the standardized vertical velocity, the stronger the ascent. The diurnal variation of total cirrus cloud cover is only significantly positive correlation with 250 hPa relative humidity at a 90% confidence level (correlation coefficient is 0.77, see Fig. 9a). It indicates that the formation of

cirrus is closely related to high-level relative humidity and this diurnal cycle correlation between cirrus and relative humidity is also found in the tropics (Chepfer et al., 2019). Chepfer et al. (2019) found that relative humidity increases by definition as the surface temperature decreases and they indicated that the joint evolution of relative humidity with cirrus is likely driven by the diurnal variation of surface temperature rather than the change in the amount of water vapour in the atmosphere.

Although there is a correlation between cloud cover of cirrus and meteorological factors, in fact, the diurnal variations of
clouds and these meteorological factors are both influenced by the diurnal variations of solar radiation. It should be noted
that the peak times of the cloud cover of different cirrus types are different to some extent, meaning that they may be
controlled by different meteorological factors and mechanisms on a diurnal scale. The correlation coefficient between the
diurnal variations of subvisible cirrus clouds and 250 hPa relative humidity reaches 0.83 (Fig. 9b), indicating that the
formation of subvisible cirrus clouds depends to large relative humidity possibly caused by cooling of water vapour at upper
troposphere, especially during nighttime (Sun et al., 2011b). As shown from the spatial distribution of the correlation
coefficient (Fig. S6b), in most areas of the TP, subvisible cirrus clouds have a positive correlation with relative humidity,
with a correlation coefficient greater than 0.3. The diurnal variation of subvisible cirrus is negative correlation with 2-m
temperature (Fig. 9b; correlation coefficient is -0.62) and 250 hPa vertical velocity (correlation coefficient is -0.36). Mace et
al. (2006) also found that optically thinner cirrus is not uncommon in regions of weak subsidence, meanwhile, thicker and
more persistent cirrus are found in large-scale ascent. There is a maximum peak in the cloud cover of thin cirrus clouds at
22:00 UTC and second peak around 15:00 UTC (Fig. 9c), and thin cirrus clouds have a weak positive correlation with
relative humidity (correlation coefficient of 0.47). From the small-scale results of each 2° grid, thin cirrus clouds have a
significant correlation with meteorological factors only in a small portion of grids (Fig. S6c, j and k).  For the opaque cirrus
cloud, its diurnal variation is positively correlated with the 250 hPa relative humidity and 250 hPa vertical velocity
(correlation coefficients of 0.59 and 0.49, respectively) (Fig. 9d). Although the correlation between opaque cirrus and 2-m
temperature doesn't pass the 90% significance test, they are found to have similar cycles with a 3-hour difference in peak
time. A few hours lag between continental convection and 2-m temperature peaks is also found over North America
(Tian et al., 2005), which attributed to a direct thermodynamic response of continental convection to the strong diurnal cycle
of 2-m temperature (Wallace, 1975). The above results indicate that, unlike subvisible cirrus, the diurnal cycle of opaque
cirrus is synchronous with diurnal variations of atmospheric convective conditions. Indeed, the diurnal cycle of deep
convection over the TP obtained by Meteosat-5 data (Devasthale and Fueglistaler, 2010) is similar with that of opaque cirrus
in our results with the peak during 10:00 UTC to 12:00 UTC (Fig. 9d). In addition, ground-based lidar measurements over
the TP also show that cirrus with optical thickness above 0.3 are always observed near deep convection (He et al., 2013).
The formations of both thick cirrus and deep convection are promoted by high 2-m temperature (Kent et al., 1995; Yang and
Slingo, 2001) and strong ascent (Mace et al., 2006; Louf et al., 2019). For cirrus, its formation is promoted by high 2-m
temperature through at least two effects (Kent et al., 1995). On the one hand, the equilibrium water vapor mixing ratio
increases with temperature based on the Clausius-Clapeyron equation, which contribute to the increase of ice water content
directly. On the other hand, the rise of temperature increases convective available potential energy (CAPE), which is
required in the transport of ice particles to the upper troposphere to form cirrus. And for deep convection, the formation is
also associated with atmospheric instability, which increases with 2-m temperature (Yang and Slingo, 2001). However, the
lag in the peak of thinner cirrus may be partly because they need time for detrainment from deep convection. The different
types of cirrus life cycles are also inferred in Feofilov and Stubenrauch (2019), which is that the cirrus anvil of deep

convection dissipates, releasing water vapour and turning to thin cirrus. Based on the above mechanisms, it is not difficult to understand that the peak times of deep convection and opaque cirrus are lagging behind that of the 2-m temperature, and peak time of subvisible cirrus is lagging behind that of deep convection and opaque cirrus. Finally, the diurnal variation of subvisible cirrus exhibits an obvious negative correlation with 2-m temperature (Fig. 9b). Of course, variation in deep convection is only one of possible mechanism to explain the correlation of diurnal cycle of opaque cirrus clouds with 2-m temperature and 250 hPa vertical velocity. One should keep in mind that cirrus formation is also influenced by other mechanisms (gravity waves, e.g.), although they may not show up on the diurnal cycle.

## 4 Conclusions and discussion

The TP, also known as the "Asian water tower", has experienced significant climate changes that are closely linked to clouds. Much of the existing research has focused on long-term changes in cloud properties. However, the diurnal variation in clouds, which plays an important role in the energy budget of the Earth-atmospheric system and climate change, has still received insufficient attention due to observation limitations. As a result, this study explores the diurnal cycle of clouds over the TP based on CATS, ISCCP, Himawari-8, ERA5, MERRA-2, and CMIP6 outputs. Related results will be helpful to improve the simulation and retrieval of total cloud cover and cloud vertical distribution, and further provide an observational constraint for simulations of diurnal cycle of surface radiation budget and precipitation over TP region. The main results are as follows:

1. The total cloud cover over the TP peaks at 06:00-09:00 UTC, and clouds are concentrated over the eastern TP. The CATS satellite can capture more clouds at night, as lidar can recognize more of the optically thin cirrus clouds that occur frequently at night than passive detectors. The largest amplitude of diurnal variations is detected by the Himawari-8 and ISCCP, but Himawari-8 significantly underestimate total cloud cover compared with CATS and ISCCP. The diurnal cycle of cloud cover from reanalysis and CMIP6 models does not show changes as dramatic as those from satellite observations.

2. Compared with the cloud vertical distribution detected by CATS, the results from ERA5 and MERRA-2 show significant underestimation of cloud cover at middle- and high- atmosphere levels. At night, there are more clouds concentrated near the surface over the southern TP according to the reanalysis results, but the CATS lidar has difficulty identifying low clouds under thick clouds. Therefore, CATS cannot obtain complete information on the diurnal variations of cloud vertical distribution over the southeastern TP, where low-level clouds are concentrated. However, CATS can still capture the pattern in cloud vertical distribution and the corresponding height of peak cloud cover at middle- and high- atmosphere levels.

3. The cloud cover of opaque cirrus clouds ($0.3< \tau <3$) dominates other cirrus cloud types over the TP and peaks at 11:00 UTC over the northeastern TP. These cirrus clouds show different characteristics from cirrus clouds in the tropics, where thin cirrus clouds dominate (Sassen et al., 2009). More thin cirrus clouds occur at night, especially in the spring.

The seasonal average cloud cover of subvisible cirrus clouds ( τ <0.03) peaks at 22:00 UTC. In particular, the cloud cover of subvisible cirrus is approximately 0.07 at night (15:00-23:00 UTC), twice as large as during daytime. But, these subvisible cirrus clouds are still difficult to detected during nighttime by using passive methods. Over the TP, the averaged cloud cover of overshooting cloud is higher at night and has a maximum value at 16:00 UTC, and its value is about 0.013, affecting the material exchange between the tropospheric and stratospheric regions.

4.    The diurnal variations of the vertically integrated divergence of moisture flux, 2-m temperature and 10-m wind speed show strong correlations with diurnal variations of total cloud cover over the TP. The diurnal cycle of subvisible cirrus clouds has a strong positive correlation with the 250 hPa relative humidity over the whole TP and show negative correlation with 2-m temperature and 250 hPa vertical velocity, which indicates that the diurnal variations of cirrus clouds are more dependent on the variations in relative humidity than the variations in other factors. However, the

diurnal variations of opaque cirrus clouds obviously correlated with 250 hPa vertical velocity and relative humidity, which shows a connection between the formation of relatively thick cirrus clouds and deep convection. The diurnal cycle of thin cirrus clouds shows a relatively weak correlation with the diurnal cycle of 250 hPa relative humidity.

The comparison of the diurnal cloud cycle between different datasets in this study suggests that large differences exist in these datasets over the TP. Indeed, compared with those of satellites, the amplitudes of cloud diurnal variations obtained

by reanalysis and CMIP6 models are too small to affect the simulation of radiation. Of course, it is impossible to completely reconcile cloud covers observed with instruments based on different observation methods. However, the part of total cloud cover difference between different datasets is possibly caused by following problems: (1) Detection sensitivity: modern passive satellites still have difficulty identifying high-level thin clouds and usually misclassify these clouds as clear sky. (2) Cloud parameterization: some cloud parameterization schemes in climate models and reanalysis data are unreasonable and

need to be further improved. (3) Discrepancies in the definitions of cloud cover between observations and models. In addition, the different temporal scales of sampling and the different quantification algorithms of cloud cover may lead to differences between satellite retrievals and model simulations (Engström et al., 2015). In recent years, many studies have contributed to reducing the uncertainty of observed and simulated cloud cover. For example, Sun et al. (2014;2015) found that using the polarization angle feature of backscattered solar radiation, super thin cirrus clouds with an optical thickness of

~0.06 can be effectively detected. This offers a new approach for detecting subvisible clouds based on low-cost passive instruments, but this new approach is also only available during the daytime. Over the TP region, our results indicate the subvisible cirrus clouds are more frequent during nighttime. It means that the detection of subvisible cirrus based on the backscattered solar radiation still cannot reduce the uncertainty of observation during nighttime. For the uncertainties in the reanalysis datasets and the other models, the accuracy of model simulations can be improved by optimizing the physical

process descriptions in parameterized schemes. The total cloud cover simulation is more realistic and directly links cloud cover to the physical processes of cloud formation (e.g., cumulus convection) instead of achieving cloud cover as a function of relative humidity and the condensate mixing ratio (Ma et al., 2018). In addition, the improvement in cloud overlap parameterization might help to optimize the simulation of cloud cover in multilayer cloud scenarios (Li et al., 2018; Li et al.,

2019). Meantime, the ground observations in regional large-scale comprehensive observation experiments greatly help to explore the mechanisms of cloud diurnal variations and to improve the model simulations (Ge et al., 2019; Yang et al., 2021).

In this study, the diurnal cycles of cirrus clouds with different optical depths are explored. Some recent studies have also further discussed the regulatory and formation mechanisms of the diurnal cycle of cirrus clouds. For example, Gasparini et al. (2019) indicate that the diurnal variations of insolation can regulate anvil cirrus evolution and radiative effects. For stratospheric cirrus over the Great Plains and surrounding areas, Zou et al. (2021) found that they are mainly developed by deep convection and gravity wave events. By using the CALIPSO observation and reanalysis dataset, Zhang et al. (2020a) point out that large-scale topographic uplift, ice particle production due to temperature fluctuations, and residuals from deep convective anvils contribute to summer cirrus cloud formation over the TP at locations under 9 km, 9-12 km, and above 12 km, respectively. However, these dynamic mechanisms of cirrus formation are complex and cannot be completely described in the diurnal cycle. Thus, although the amount of cirrus cloud cover over the TP is related to meteorological factors from the perspective of microphysics, the present analysis still cannot provide a clear interpretation of which mechanisms drive the diurnal cycles of cirrus clouds of different optical depths. In addition, some previous studies have indicated that diurnal cycles of cloud properties (e.g., cloud droplet size and cloud liquid water path) is related with the variation of aerosol loading in their study periods (Matsui et al., 2006; Ntwali and Chen, 2018), but these studies didn't address the impact of meteorological factors on the diurnal cycle of cirrus clouds. By using the 33 months of dust aerosols extinction coefficient and meteorological factors, Wang et al. (2022) showed a robust dependence of diurnal cycle of supercooled water cloud cover on the variation of dust aerosol extinction coefficient instead of other dust load indicators and meteorological parameter. These results demonstrate that the aerosol loading can affect the diurnal cycle of cloud cover, however, whether aerosol loading over the TP region is the major driving factor of diurnal cycle of cirrus clouds is still unclear. Thus, future works also should pay more attentions on the impact of aerosol on the diurnal cycle of cirrus cloud over the TP region.

**Appendix A: The comparison of cloud spatial distribution between different datasets and CATS every 3 hours**

To further quantify the spatial consistency of total cloud cover from passive satellites, reanalyses and models with CATS observations, the Taylor diagram (Taylor, 2001) is used to provide the standard deviation and centred root-mean-square deviation (RMSD, purple circle) normalized by the observed values and the spatial correlation coefficients between other datasets and CATS observations (Fig. A1). Here, all datasets are uniformly interpolated into 2° by 2°. The standard deviation and RMSD show the changes in amplitudes and phases of the different datasets, respectively. That is, the closer the standard deviation shown by the points in Fig. A1 is to the red line, the closer the amplitude of the spatial difference of the total cloud cover of the datasets and CATS. The smaller the RMSD is, the more similar the distribution pattern is to that of CATS. The results show that the correlation coefficient between CMIP6 multimodel mean and CATS is large with an average of 0.47, but the spatial consistency between certain CMIP6 models (e.g., IPSL-CM6A-LR) and CATS is poor. The ISCCP, which is the closest dataset to the regional mean total cloud cover of CATS in Fig. 2, shows a weaker correlated with

CATS in spatial distribution every 3 hours with an average of 0.23. The standard deviation in the Himawari-8 is relatively large compared to other datasets from 00:00 UTC to 06:00 UTC, which indicates that the Himawari-8 overestimates the spatial difference when the mean total cloud cover is relatively low. ERA5 has a smaller standard deviation than CATS during 03:00~12:00 UTC. The minimum correlation coefficient between ERA5 and CATS is 0.25 at 09:00 UTC and the average correlation coefficient is 0.50. Despite the different amplitudes of diurnal variation in total cloud cover between CATS and MERRA-2, their spatial distribution shows high consistency, with a largest average correlation coefficient among all datasets of 0.57. The standard deviation of MERRA-2 is less than that of CATS during the whole day.

**Data availability**

CATS datasets are downloaded from the NASA Langley Research Center Atmospheric Science Data Center (ASDC) website: https://eosweb.larc.nasa.gov/project/CATS-ISS?level=2. The ISCCP data are available from website: https://www.ncei.noaa.gov/products/international-satellite-cloud-climatology. The Himawari-8 data are available from the Japan Aerospace Exploration Agency, Earth Observation Research Center (JAXA/EORC) website: https://www.eorc.jaxa.jp/ptree/. The ERA5 datasets are available from Climate Data Store (CDS) website: https://cds.climate.copernicus.eu/cdsapp#!/home. The MERRA-2 datasets are available from the Global Modeling and Assimilation Office (GMAO) website: https://gmao.gsfc.nasa.gov/reanalysis/MERRA-2/. The CMIP6 outputs are downloaded from the Earth System Grid Federation (ESGF) website: https://esgf-node.llnl.gov/search/cmip6/.

**Competing interests**

Some authors are members of the editorial board of journal Atmospheric Chemistry and Physics. The peer-review process was guided by an independent editor, and the authors have also no other competing interests to declare.

**Author contribution**

YZ and JL organized the paper and performed related analysis. YZ prepared the manuscript with contributions from all co-authors. JL conceptualized the paper and revised the whole manuscript. LZ, CD and YL downloaded the data used in this study. YH, BJ and JH provided suggestions for this study. All authors contributed to the discussion of the results and reviewed the manuscript.

**Acknowledgements**

This research was jointly supported by the Strategic Priority Research Program of the Chinese Academy of Sciences (XDA2006010301) and the National Science Fund for Excellent Young Scholars (42022037). We would like to thank the

CATS, ISCCP, Himawari-8, ERA5, MERRA-2 and CMIP6 science teams for providing excellent and accessible data
products that made this study possible.

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

1065

| Data | Temporal Coverage | Spatial Resolution | Temporal Resolution |
|------|------------------|-------------------|---------------------|
| CATS | 2015.3.26-2017.10.29 | orbital profiles | orbital profiles |
| ISCCP | 2015.3-2017.6 | $1° \times 1°$ | 3 hours |
| Himawari-8 | 2016.1.1-2017.10.29 | $0.05° \times 0.05°$ | 10 min |
| ERA5 | 2015.3.26-2017.10.29 | $0.25° \times 0.25°$ | 1 hour |
| MERRA2 | 2015.3.26-2017.10.29 | $0.5° \times 0.625°$ | 1 hour for single level/3 hours for pressure level |

| The CMIP6 model names and their hotizontal resolutions | | | |
|------|------|------|------|
| No | Source ID | Resolution | Temporal Resolution |
| 1 | ACCESS-CM2 | 144×192 | 3 hours |
| 2 | AWI-CM-1-1-MR | 192×384 | 3 hours |
| 3 | BCC-CSM2-MR | 160×320 | 3 hours |
| 4 | CMCC-CM2-SR5 | 192×288 | 3 hours |
| 5 | CMCC-ESM2 | 192×288 | 3 hours |

| | | | |
|---|---|---|---|
| 6 | EC-Earth3 | 256×512 | 3 hours |
| 7 | EC-Earth3-Veg | 256×512 | 3 hours |
| 8 | IITM-ESM | 94×192 | 3 hours |
| 9 | IPSL-CM6A-LR | 143×144 | 3 hours |
| 10 | KACE-1-0-G | 144×192 | 3 hours |
| 11 | KIOST-ESM | 96×192 | 3 hours |
| 12 | MIROC6 | 128×256 | 3 hours |
| 13 | MPI-ESM1-2-HR | 192×384 | 3 hours |
| 14 | MPI-ESM1-2-LR | 96×192 | 3 hours |
| 15 | MRI-ESM2-0 | 160×320 | 3 hours |
| 16 | NESM3 | 96×192 | 3 hours |
| 17 | TaiESM1 | 192×288 | 3 hours |

**Table 1: The temporal coverage and resolution of datasets used in this study. The temporal coverage of all CMIP6 model outputs is from March 26, 2015 to October 29, 2017.**

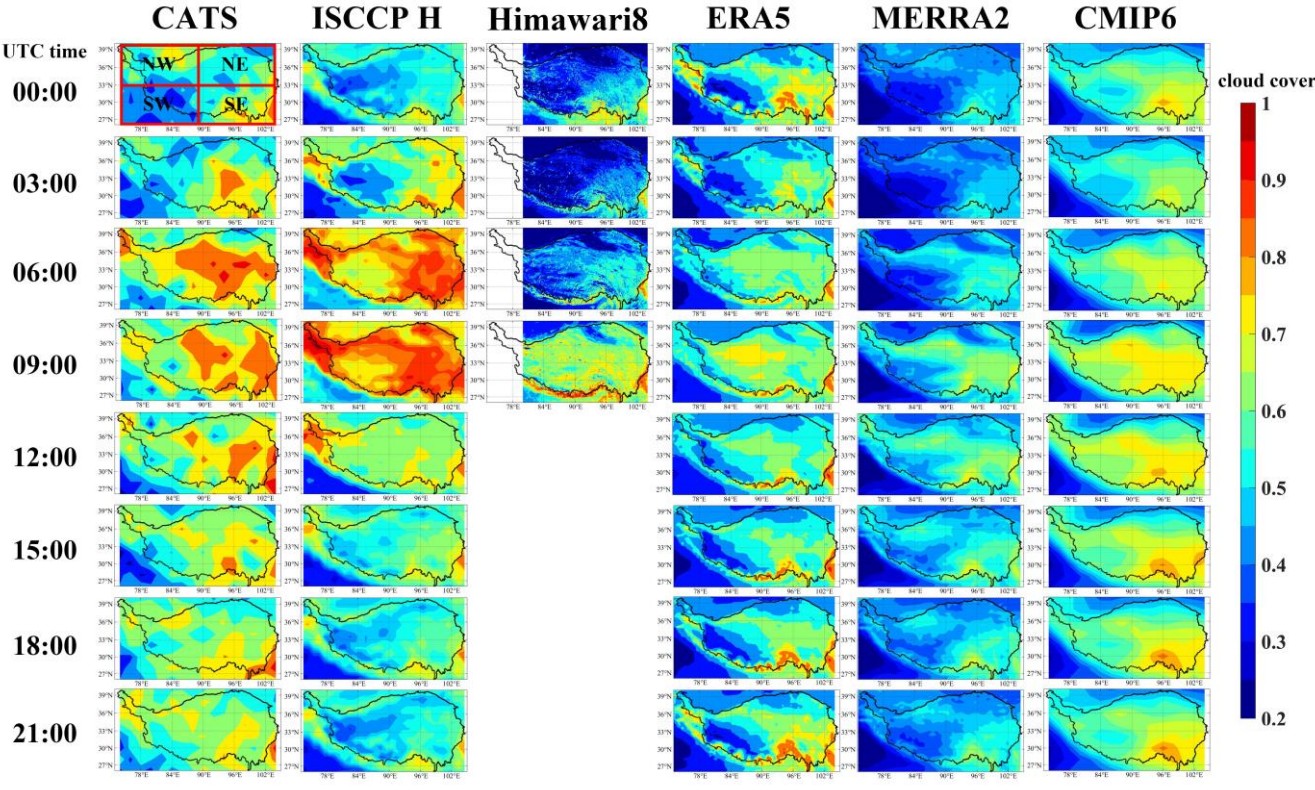

1070

Figure 1: The spatial distribution of 3-hourly averaged total cloud cover over the TP based on CATS, ISCCP-H, Himawari-8, ERA5, MERRA-2, CMIP6 multimodel mean. The different TP regions involved in the following part of this study are shown in the upper-left image.

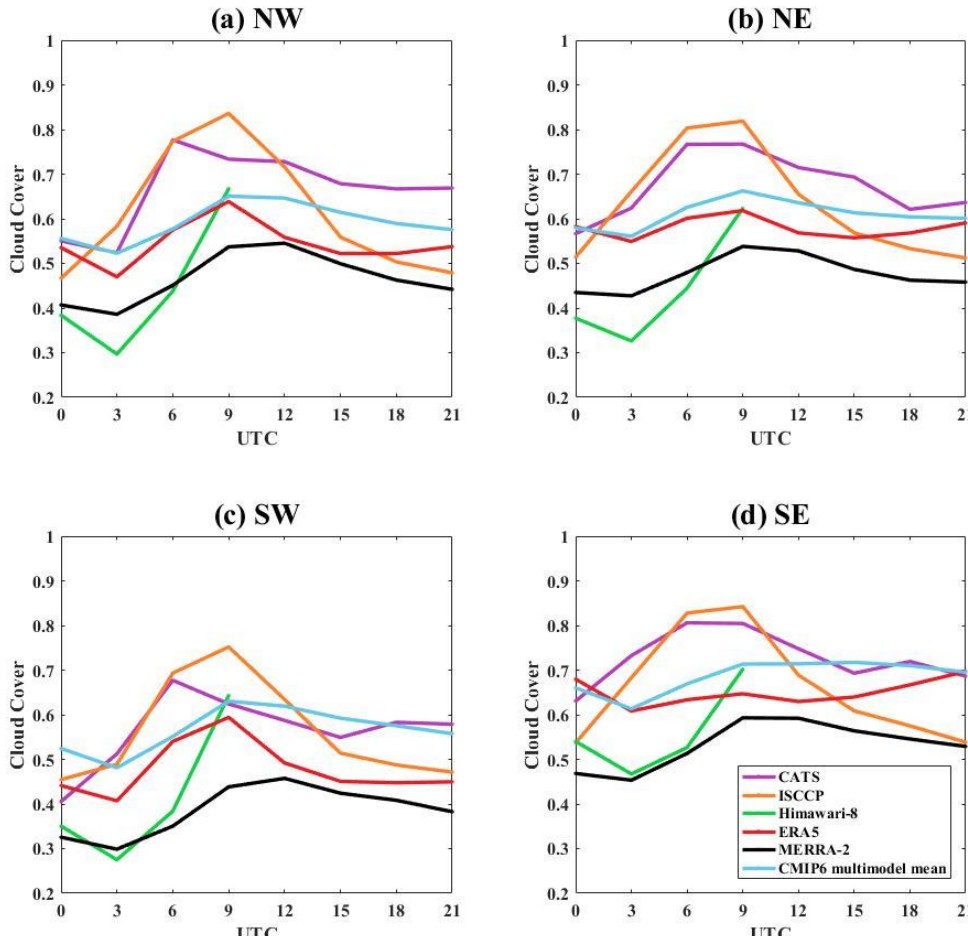

Figure 2: The 3-hourly mean total cloud cover in different regions of TP based on CATS (purple lines), ISCCP-H (orange lines), Himawari-8 (green lines), ERA5 (red lines), MERRA-2 (black lines), CMIP6 multimodel mean (blue lines). (a) The northwestern TP (b) The northeastern TP (c) The southwestern TP (d) The southeastern TP. The regions are divided by latitude and longitude lines of 33°N and 89°E and the boundary of TP (shown in Fig. 1).

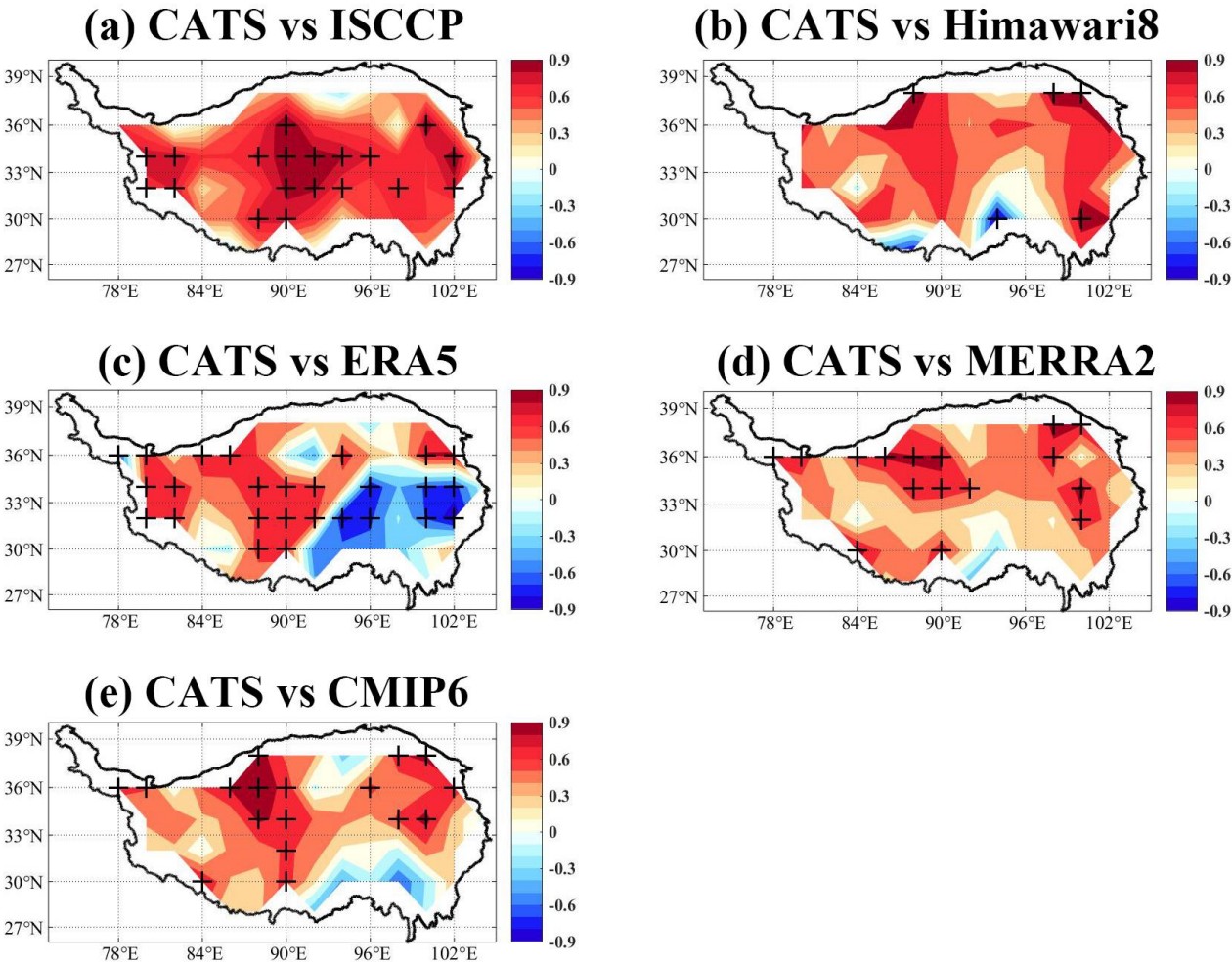

Figure 3: The spatial distribution of correlation coefficients of diurnal cycle for total cloud cover between CATS and other datasets. The grids are marked with "+" if the correlation at these grids pass the significance test by 90%. Only cloud cover results at 00:00 UTC, 03:00 UTC, 06:00 UTC, 09:00 UTC are used in the correlation between CATS and Himawari-8, as only daytime cloud cover is available from Himawari-8.

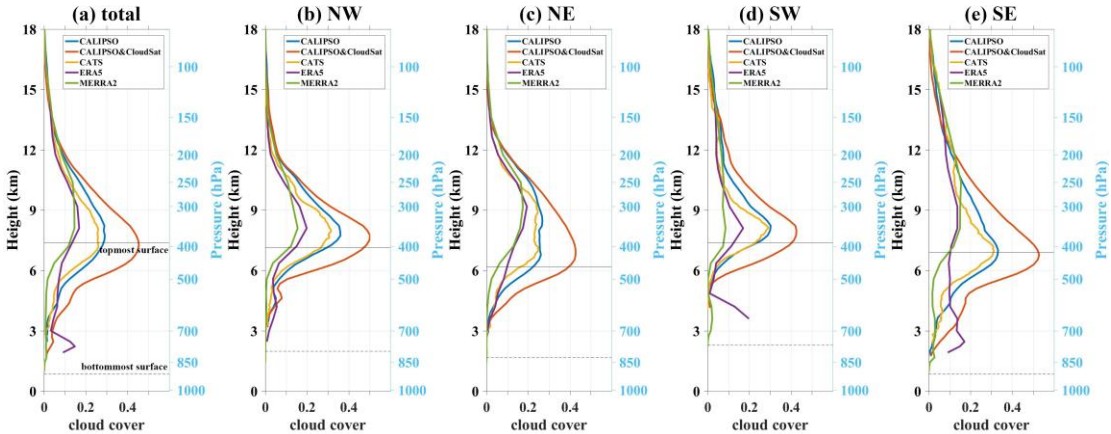

Figure 4: The cloud vertical distribution in different regions of the TP based on CALIPSO (blue lines), 2B-GEOPROF-lidar (CALIPSO&CloudSat, red lines), CATS (yellow lines), ERA5 (purple lines), MERRA-2 (green lines) at the hour closest to the CloudSat and CALIPSO daytime overpass time. (a) The whole TP (b) The northwestern TP (c) The northeastern TP (d) The southwestern TP (e) The southeastern TP. The regions are divided by latitude and longitude lines of 33° N and 89° E and the boundary of TP (shown in Fig. 1). The height here represents the height above the mean sea level. The horizontal solid black lines represent the topmost surface altitude and the dashed black lines represents the bottommost surface altitude obtained in CATS DEM elevation.

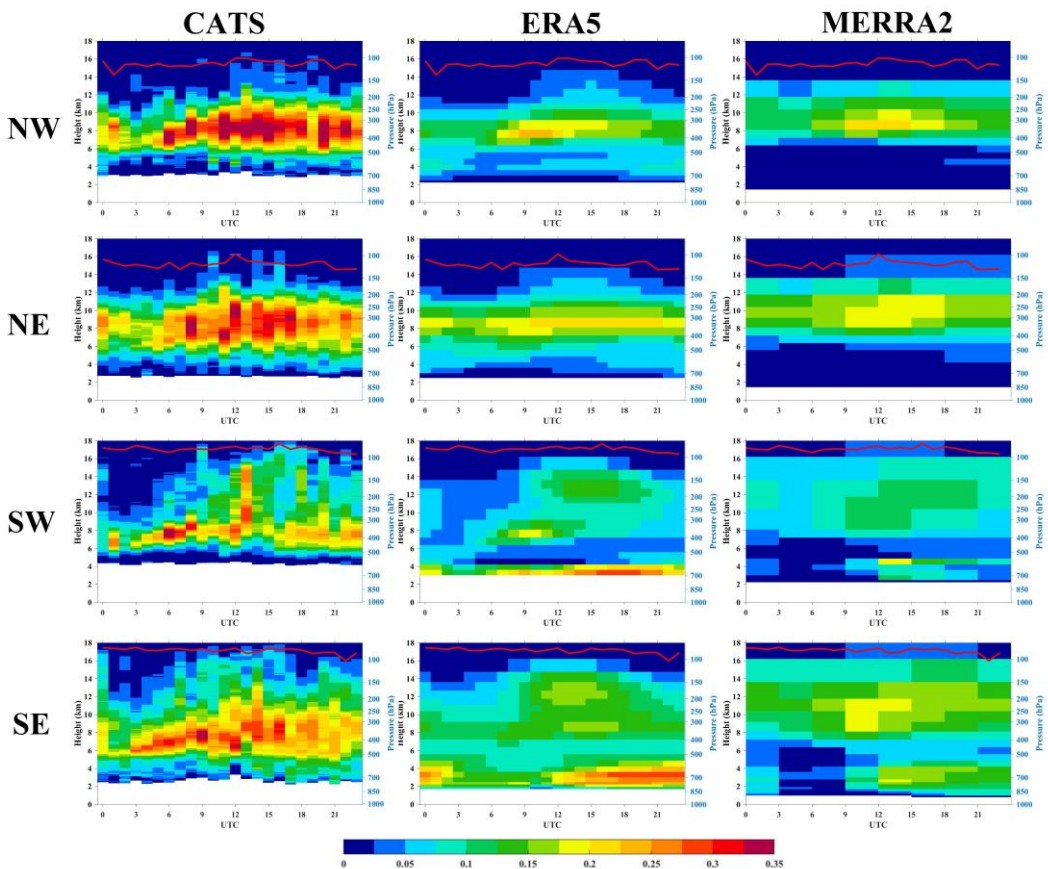

**Figure 5: The hourly vertical distribution of cloud cover over different regions of TP based on CATS, ERA5, MERRA-2. The red lines represent the tropopause height. The first to fourth lines represent the results over the northwestern TP, the northeastern TP, the southwestern TP and the southeastern TP, respectively. The regions are divided by latitude and longitude lines of 33° N and 89° E (shown in Fig. 1). The grids with total sample number less than 50 are blank.**

1095

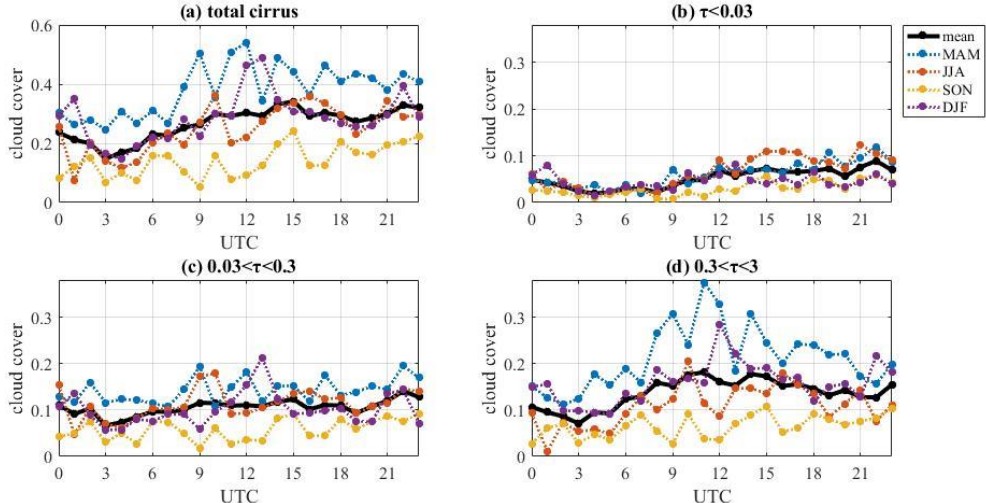

**Figure 6: The hourly cloud cover of different types of cirrus of different seasons. The black lines represent the results of annual average, the blue lines represent the results of spring, the red lines represent the results of summer, the yellow lines represent the results of autumn, the purple lines represent the results of winter. All seasons here are northern hemisphere seasons. (a) represents the cloud cover of all cirrus. (b) represents the subvisible cirrus (optical thickness less than 0.03). (c) represents the thin cirrus (optical thickness between 0.03 and 0.3). (d) represents the opaque cirrus (optical thickness between 0.3 and 3).**

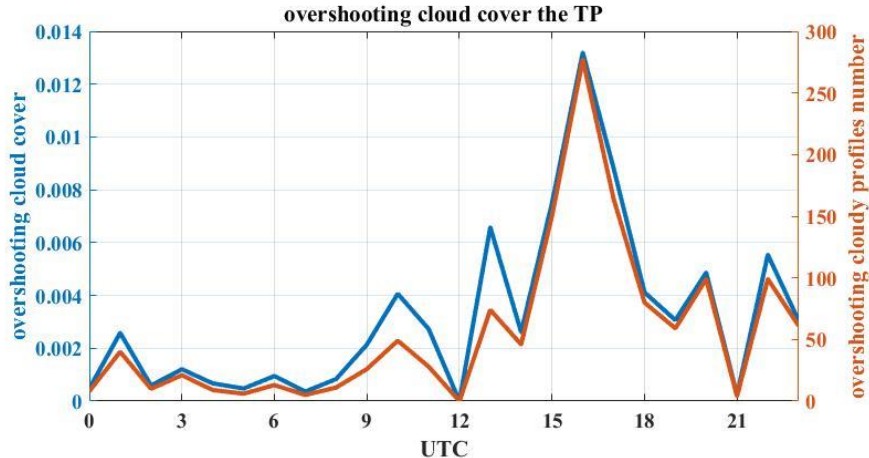

**Figure 7: The hourly cloud cover of cirrus shooting over tropopause based on CATS over the TP (blue line). The number of overshooting cloudy profiles every hour (red lines).**

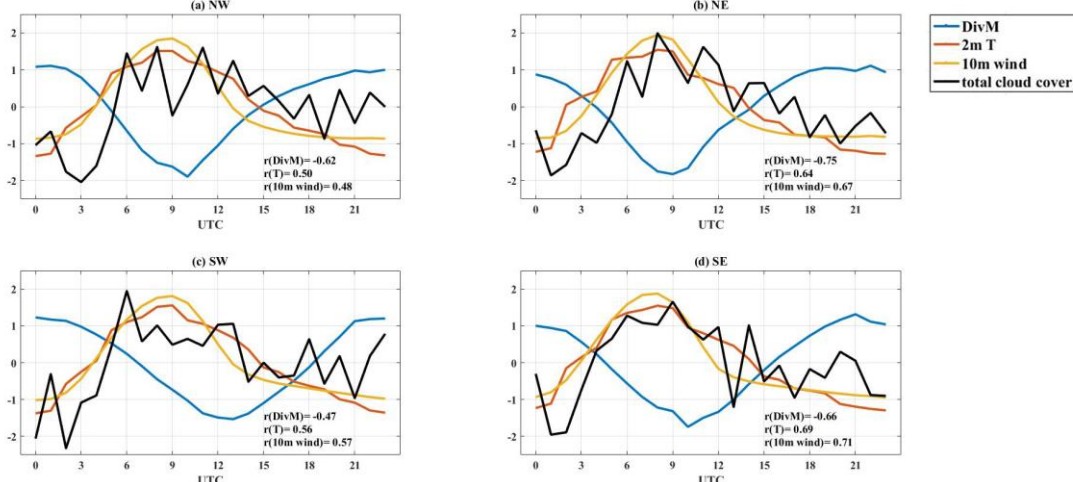

**Figure 8: The standardized total cloud cover (black lines) and vertically integrated divergence of moisture flux (kg/m$^{-2}$s$^{-1}$) (blue lines), 2-m temperature (K) (red lines), 10-m wind speed (m/s) (yellow lines) of the different regions of TP. (a) The northwestern TP (b) The northeastern TP (c) The southwestern TP (d) The southeastern TP. The correlation coefficients are indicated in the bottom right corner. The correlation coefficient in bold indicates that it can pass the 90% significance test.**

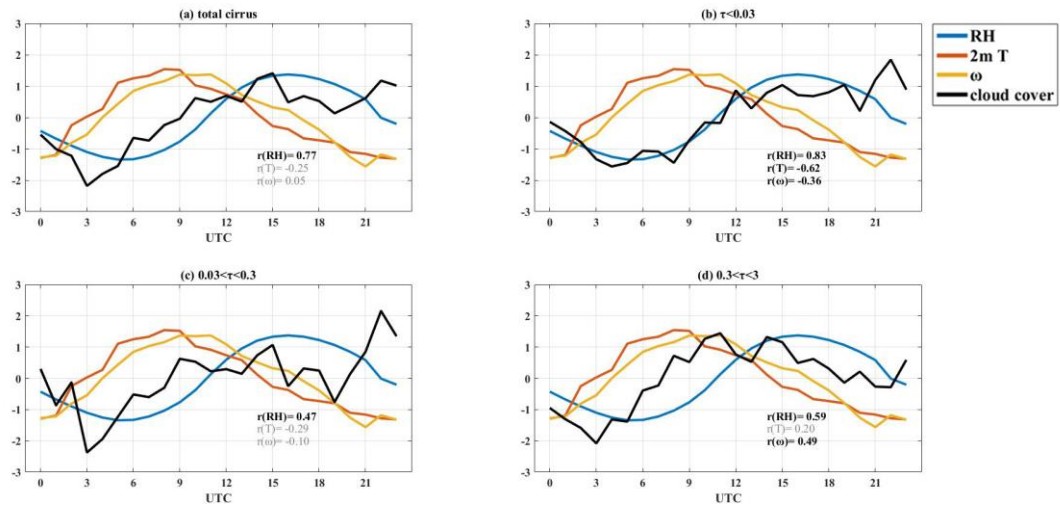

**Figure 9: The standardized cloud cover of different types of cirrus (black lines) and 250 hPa relative humidity (%) (blue lines), 2-m temperature (K) (red lines), 250 hPa vertical velocity (Pa/s) (yellow lines) over the TP. (a) all cirrus. (b) the subvisible cirrus (optical thickness less than 0.03). (c) the thin cirrus (optical thickness between 0.03 and 0.3). (d) the opaque cirrus (optical thickness between 0.3 and 3). The correlation coefficients are indicated in the bottom right corner. The correlation coefficient in bold indicates that it can pass the 90% significance test.**

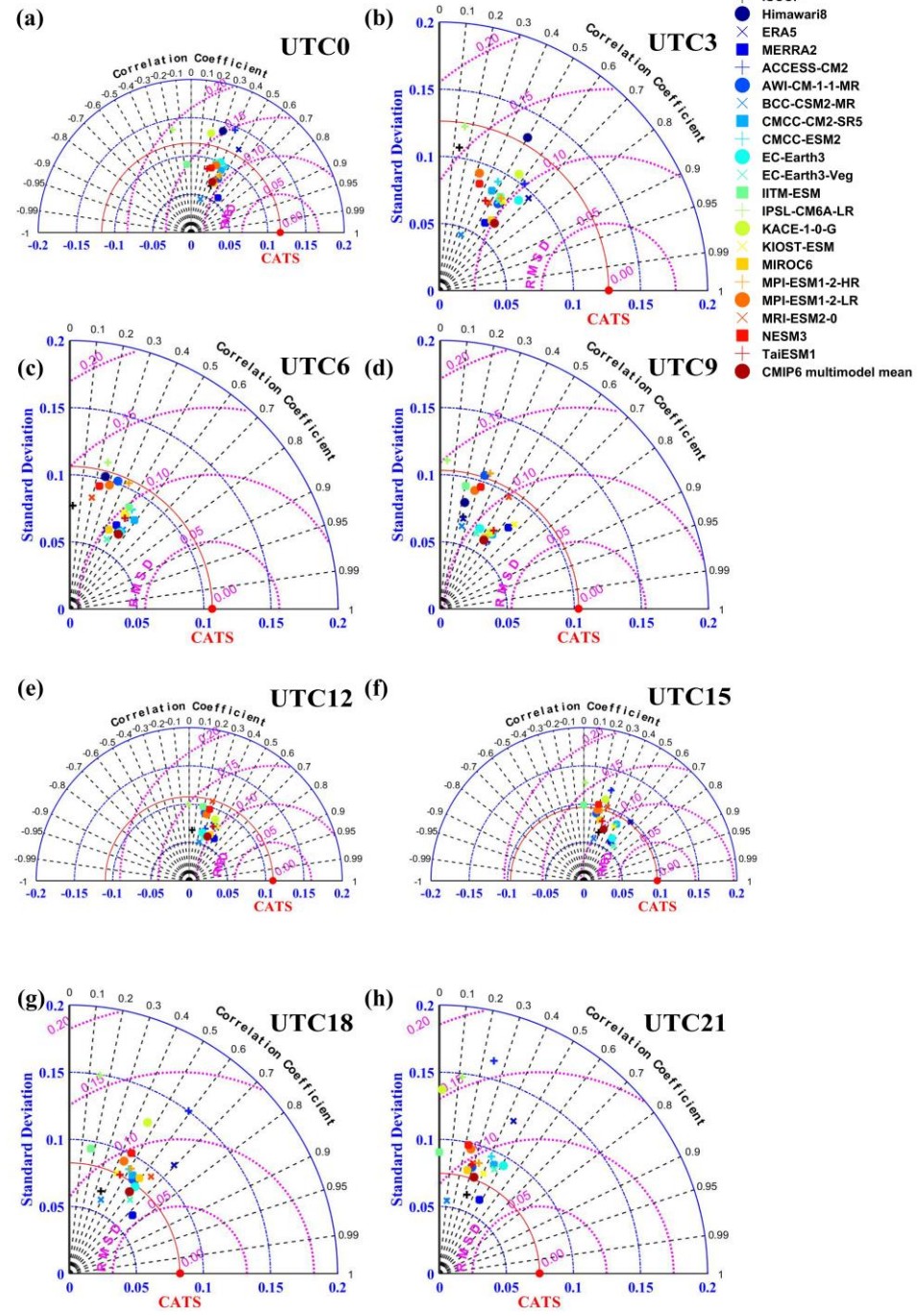

**Figure A1: The Taylor diagram describing spatial consistency of total cloud cover between different datasets and CATS every 3 hours. The distance from the origin of the coordinate axis represents the standard deviation for each dataset in spatial distribution. The distance from the red dot labelled 'CATS' represents centred root-mean-square deviation (RMSD, purple circle). The time represented by each image is displayed in the upper-right corner, and the time here is local time.**