# Peer review of "Diurnal cycles of cloud cover and its vertical distribution over the Tibetan Plateau revealed by satellite observations, reanalysis datasets and CMIP6 outputs"

_Atmospheric Chemistry and Physics, 2022_

## Author Comment (AC1)

**Response to Reviewer #1's Comments:**

Yuxin Zhao et al. (Author)

**We are very grateful for the Reviewer #1' detailed comments and suggestions, which help us improve this paper significantly. Based on the suggestions from reviewer, we adjust the related calculation method of cloud vertical distribution from CATS and CALIPSO profiles. In addition, the CMIP6 historical runs are replaced by SSP5-8.5 runs as suggested. In particular, we also correct some bugs in the data processing, and revise the corresponding figures and descriptions. Based on the comments and suggestions, we also correct inappropriate or unclear descriptions in the manuscript.**

**Please see our point-to-point reply to comments. All revisions were shown in revised manuscript by using track changes.**

**General responses:**

1.  Major point: My main concern with the paper is a relatively small one: I think there might be a problem in the normalization of CATS (and CALIOP in Section 3.2) cloud detections by the number of profiles. To calculate the cloud fraction at a given altitude level, profiles must be normalized by the actual number of profiles that were able to sample the atmosphere at that altitude level. Profiles that are fully attenuated above that altitude should be removed from the number of profiles used for normalization. In other words, you cannot consider the number of profiles to be constant over a given vertical column: the number of profiles sampled by CATS should be always altitude-dependent. For each profile you should check if it is opaque or not (Percent_Opacity_Fore_FOV variable in CATS Layer products). If it is not opaque, that profile contributes to the number of profiles at all altitude levels. If that profile is opaque, you should not count that profile in the total number of profiles for all altitude levels below the base of the lowest cloud layer detected in that profile. If you do not consider altitude-dependent number of profiles in the way I've described above, at low altitudes the cloud cover will be underestimated, as you take into account profiles that are fully attenuated, in which clouds cannot possibly be identified. If that is indeed the way you have processed the CATS data, please make it explicit in the text. If not, please revise your analysis and results and their discussion. Such a revision would mainly affect the low-altitude results and not the cirrus results.

**Response:** We agree with the reviewer and are very appreciated for reviewer providing such helpful comments and suggestions. Indeed, the impact of opacity of the CATS profiles on the calculated cloud cover profile has been not considered in our previous manuscript, it will not affect the total cloud cover but really underestimate the cloud cover at low-altitude. Following the suggestion from reviewer, therefore, we remove the samples below the opaque layers when calculating vertical cloud fraction, and revise the related figures, results and their discussions. The Fig. R1 shows the difference of cloud vertical distribution between before and after removing the profiles that are fully attenuated below opaque layers. We can see that the underestimation of cloud cover mainly occurs at low altitude (from 3km to 9km), and the maximum biases are around 7km and even exceed 0.04 regardless of CATS or CALIPSO. After correcting this statistical error, we redraw the Fig. 4, Fig. 5 and corresponding supporting information (see Fig.S1). The Fig. R2 (same as the Fig.4 in the revised manuscript) indicates that the cloud cover between 3km to 9km slightly increases compared with our previous results. In addition, we also correct

some bugs in data processing, it results in a little bit change of pattern of cloud vertical profile from these datasets than those of previous version. The corrects include: (1) Removing the profiles that are fully attenuated below opaque layers for CATS and CALIPSO; (2) 2B-GEOPROF-LIDAR product is only released till the end of 2016 at the time of writing, but now we can extend the observation period of 2B-GEOPROF-LIDAR dataset and CALIPSO observation to the Oct./2017, and keep consistency with that of CATS. (3) Correcting the bug in pressure-height converting for MERRA2 and ERA5. (4) In the previous version, the CATS L2O 5 km profile product, which is used to provide the tropopause height information (see the Comment #47), have some loss at observation period due to link issue, but now we already fill the miss data.

[Figure]

**Figure R1:** the difference of cloud vertical distribution between before and after removing the profiles that are fully attenuated below opaque layers from CALIPSO (blue lines) and CATS (red lines) at the hour closest to the CloudSat and CALIPSO daytime overpass time. (a) The whole TP (b) The northwestern TP (c) The northeastern TP (d) The southwestern TP (e) The southeastern TP. The regions are divided by latitude and longitude lines of 33°N and 89°E (shown in Fig. 1).

[Figure]

**Figure R2 (same as the Fig. 4 in the revised manuscript):** The cloud vertical distribution in different regions of the TP based on CALIPSO (blue lines), 2B-GEOPROF-lidar (CALIPSO&CloudSat, red lines), CATS (yellow lines), ERA5 (purple lines), MERRA-2 (green lines) at the hour closest to the CloudSat and CALIPSO daytime overpass time. (a) The whole TP (b) The northwestern TP (c) The northeastern TP (d) The southwestern TP (e) The southeastern TP. The regions are divided by latitude and longitude lines of 33°N and 89°E and the boundary of TP (shown in Fig. 1). The height here represents the height above the mean sea level. The horizontal solid black lines represents the topmost surface altitude and the dashed black lines represents the bottommost surface altitude

obtained in CATS DEM elevation.

Fig. R3 replaces Fig. 5 in Section 3.2 in the revised manuscript. Similar, after above corrections, the cloud cover shown in Fig. R3 is larger than those shown in Fig. 5 of previous manuscript, especially for low-level clouds over the southeastern TP. It can be found that there are more clouds below 4km (the cloud cover is around 0.15) from 15:00 UTC to 21:00 UTC in the southeastern TP. Besides, the peak value of cloud cover is also slightly increased as stated in the Fig. R1. In summary, by removing the profiles that are fully attenuated below opaque layers in the total samples and correcting some bugs, our results have a little variation, but their changes are weak and thus will not affect our main conclusions. In the revised manuscript, the description in Section 3.2 and Section 4 involving the vertical distribution of cloud cover will be changed accordingly the changes of Fig. 4 and Fig. 5, respectively. In addition, related information is updated and we also add a short description about this processing in the section 2.1: "But it is worth noting that we also use the parameter "Percent_Opacity_Fore_FOV" in the CATS layer product to check the opacity of each profile. If it is not opaque, that profile contributes to the number of profiles at all altitude levels. If that profile is opaque, we don't count that profile in the total number of profiles for those altitude levels below the base of the lowest cloud layer detected in that profile. For CATS, a profile is considered opaque if no surface return is detected in all level 1 (L1B) 350 m profiles that make up that L2O 5 km profile." (See Line: 134-139, in the revised manuscript)

[Figure]

**Figure R3 (same as the Fig. 5 in the revised manuscript):** The hourly vertical distribution of cloud cover over different regions of TP based on CATS, ERA5, MERRA-2. The red lines represent the tropopause height. The first to fourth lines represent the results over the northwestern TP, the northeastern TP, the southwestern TP and the southeastern TP, respectively. The regions are divided by latitude and longitude lines of 33°N and 89°E (shown in

Fig. 1). The grids with total sample number less than 50 are blank.

2. Minor points: l. 20-21: "we find that cirrus clouds... show significant... spatial and temporal distribution characteristics" -- what this means is unclear to me. Please rephrase.
**Response:** We change this sentence to "We further find that cirrus clouds, which are widespread over the TP, show significant diurnal variations with averaged peak cloud cover over 0.35 at 15:00UTC." (See the Line: 21-22 in the revised manuscript).

3. l. 22: "Be different from tropic" -- do you mean "Unlike in the tropics"?
**Response:** Yes, we mean "Unlike in the tropics". We are sorry for the confusion. It is corrected in the revised manuscript. (See the Line: 22).

4. l. 23-24: "The cloud cover... are influenced"
**Response:** It is corrected in the revised manuscript. (See the Line: 24).

5. l. 32 and many others: "convection activities" this should be described as "convective activity" (singular).
**Response:** Similar errors have been corrected in the revised manuscript. (See the Line: 36).

6. l. 60: "ERA-20C" this is not defined.
**Response:** Thanks for your comment! The full name "ECMWF's first atmospheric reanalysis of the 20th Century" is added to the revised manuscript. (See the Line: 65).

7. l. 69: please check the order of references here and throughout the paper. Here Zou et al. 2020 should be cited last.
**Response:** Thank you for pointing out this issue. The order of references here and throughout the paper are corrected in revised manuscript.

8. l. 80: "detect": "document" would be better.
**Response:** We agree with the reviewer. It is corrected in the revised manuscript. (See the Line: 87).

9. l. 83-87: this is a very long sentence. Please consider ways to split it up.
**Response:** This long sentence is split in the revised manuscript.

10. l. 86: as written, it looks like Yorks et al. 2016 talks about CALIOP data, which is not the case.
**Response:** We are sorry to mislead the reviewer. This sentence is corrected in the revised manuscript. (See the Line: 95-97).

11. l. 90: "this makes it possible for the CATS to analyze..." CATS does not analyze, CATS is an instrument. A scientist can analyze CATS data.
**Response:** We agree with the reviewer. It is corrected in the revised manuscript. (See the Line: 100).

12. l. 90 and throughout: instead of "the CATS" and "the CALIOP", please use "CATS" and "CALIOP" instead.

**Response:** They are corrected throughout in revised manuscript.

13. l. 108: "to clarify the cloud/aerosol layer and retrieve its properties": this is confusingly written, please revise.

**Response:** It is corrected to "CATS employs a similar atmospheric layer-detection algorithm as CALIOP to identify the cloud/aerosol layer and retrieve layer properties (i.e., layer height and thickness, optical depth et al.)". (See the Line: 119-121).

14. l. 116: "rotation" do you mean "operation"?

**Response:** Yes, here we mean "operation". It is corrected in revised manuscript. (See the Line: 129).

15. l. 122: please see main comment #1.

**Response:** Thanks for your comment! Based on the suggestion, we revise the method and related figures, meantime, we also add a short description about this processing in the section 2.1: "But, it is worth noting that we also use the parameter "Percent_Opacity_Fore_FOV" in the CATS layer product to check the opacity of each profile. If it is not opaque, that profile contributes to the number of profiles at all altitude levels. If that profile is opaque, we don't count that profile in the total number of profiles for those altitude levels below the base of the lowest cloud layer detected in that profile. For CATS, a profile is considered opaque if no surface return is detected in all level 1 (L1B) 350 m profiles that make up that L2O 5 km profile." (See Line: 134-139 in the revised manuscript).

16. l. 146: "the detection range of the AHI moves daily": unclear, does this mean the detection range changes from one day to the next? How is this "range" defined? The range of what? It is unclear to me what was your intention when including Himawari-8 imagery into the comparison -- the results show quite clearly that its retrieved cloud covers are less robust than the other datasets you've considered. The strength of the AHI imagery appears to be its very high horizontal resolution compared to the other datasets, is that the reason for its inclusion? Do you think the insights it provides justify its inclusion?

**Response:** We are sorry to mislead the reviewer. In fact, the detection range of the AHI is fixed due to Himawari-8 is a geostationary satellite. At some hours, the field of view of AHI also includes some pixels without cloud mask (shown in data quality assurance (QA) flag: cloud retrieval algorithm flag) at solar zenith angles above 80°. It results in the cloud covers at these pixels are lack because the cloud dataset during night-time has not been released at the time of writing. In the revised manuscript, we correct related description as: "In addition, until now, only cloud cover from AHI during daytime is available, thus we merely consider the period in which there are complete data over the TP, which is during 00:00 UTC to 10:00 UTC". Indeed, the strength of the AHI imagery is its very high horizontal resolution compared to the other datasets. Here, we include Himawari-8 imagery into the comparison is because the diurnal cycle study of total cloud cover over the whole TP region almost are based on the ISCCP or Himawari-8 AHI (Lei et al., 2020; Shang et al., 2018). Thus, for ease of comparison with previous studies, we include the Himawari-8 imagery into the comparison.

17. l. 153: "unlike satellite observations... cloud characteristics from reanalysis data largely depend on atmospheric numerical models and data assimilation schemes" true, but data assimilation is the process by which observations (including satellite) are taken into account in reanalysis datasets.

**Response:** We agree with reviewer. The "Unlike satellite observations" is not an appropriate expression

here, it is deleted from the revised manuscript.

18. l. 155 you specify the number of vertical levels for MERRA-2, please do the same for ERA5.
**Response:** It is added in revised manuscript. (See the Line: 173).

19. l. 166: if the dimensions include latitude, longitude, height and time, then it is a 4-dimensional dataset.
**Response:** Yes, it is corrected in revised manuscript (See the Line: 184-186).

20. l. 169: I understand that in what follows you've averaged the cloud covers from these 12 models. CMIP6 includes more than 12 models. How were these specific 12 models selected for your particular study? How does this selection affect your results? How did you reconcile model outputs that were on different spatial grids? What did you select as the main grid?
**Response:** Thanks for your comment! In our study, the main reason that we selected these models is because only the 3-hourly cloud cover outputs from these 12 models are available at the time of writing. For the spatial resolution of models, in the revised manuscript, we uniformly linearly interpolate all model outputs to $2° \times 2°$ grid instead of $0.5° \times 0.5°$ in previous manuscript in order to better compare with that from CATS. In the revised manuscript, we add the related description at Line: 200-202 as: "In the subsequent analysis, all model outputs, reanalysis, ISCCP and Himawari-8 are uniformly linearly interpolated to the $2° \times 2°$ grid after analyzing Fig.1 to keep consistency with CATS observation". It is worth noting that we have reselected the model outputs based on the next comment from reviewer. That is, using the cloud cover simulation under RCP8.5 condition during 2015-2017 period instead of the historical runs over 1979-2014. Detailed information, please see the next response.

21. l. 172: you mean the CMIP6 historical runs stop at 2014, right? Couldn't you use RCP8.5 runs? (their emission scenarios follow quite closely the actual emissions). They might cover the 2015-2017 period. Otherwise you are comparing satellite observations over 2015-2017 (i.e. two rather recent years) with an average over 1979-2014 (i.e. 33 more years). The observations will likely be much more affected by climate change than the model output. If you can't use RCP8.5 output, please address this somehow in the text.
**Response:** We very thank reviewer for providing this important suggestion. Indeed, the CMIP6 historical runs are stop at 2014. Based on the suggestion from reviewer, we replace the historical outputs with SSP5-8.5 runs, which is an upgrade of RCP8.5. For the RCP8.5 runs, there are 17 CMIP6 models are available for the 3-hourly cloud cover output during the 2015-2017 period. Thus, in the revised manuscript, all figures and descriptions involving CMIP6 outputs are changed, about the detailed information, please see the Section 2.5 and Section 3.1 in the revised manuscript.

22. l. 174: Unclear. The paragraph opens by saying that you use the 3-hourly cloud area fraction from 12 CMIP6 models, and now it says the CMIP6 simulations are unavailable for the 3-hourly cloud area fraction? How can you use that data is it is unavailable?
**Response:** We are sorry to mislead the reviewer. In fact, our mean is that the cloud area fraction from CMIP6 models with lidar simulator (e.g., CATS simulator) is unavailable for the 3-hourly resolution. Thus, we use the 3-hourly cloud area fraction from CMIP6 models without lidar simulator. We already revise the related sentence.

23. l. 176: "Table 1..." please make this a new paragraph.
**Response:** It is changed in the revised manuscript. (See the Line: 203-203).

24. l. 181: Are results shown in Figure 1 aggregated over the entire time periods? Please make this explicit.
**Response:** Yes, the results in Fig.1 are aggregated over the entire time periods. We add the related explanation in the revised manuscript. (See the Line: 210-211).

25. l. 187: "especially the total cloud cover": Figure 1 shows only the total cloud cover. Especially compared to what?
**Response:** This is a confusing expression. It is corrected in the revised manuscript

26. l. 199: "CERE"
**Response:** It is corrected to "CERES" in revised manuscript.

27. l.213: "The cloud cover from MERRA-2 is lowest": Himawari cloud cover is lower overall
**Response:** We correct some bugs in the results and figures, and related discussions are revised.

28. l. 215: "except in the ITCZ": this is not relevant here
**Response:** We agree with reviewer! Based your comment, we delete it in the revised manuscript.

29. l. 221: "cannot be overlapped": cannot overlap
**Response:** Related words are revised in revised manuscript.

30. l. 222: this sentence is not useful.
**Response:** It is deleted in revised manuscript.

31. l. 229: why do you start by describing the southwestern TP (Fig. 2c)? You don't say a lot about this figure and very quickly switch to northwest TP. It seems to me most of the discussion of northwest TP applies equally to southwest TP. What logic drives the order in which you discuss the figures?
**Response:** We are sorry to mislead the reviewer. Indeed, the logic of discussion about this figure is disordered in the previous manuscript. In the revised manuscript, we reconstruct this sentence. (See the Line: 273-300).

32. l. 231-233: the same can be said from cloud covers in the southwestern TP.
Response: In the revised manuscript, we correct some errors in the figures and reconstruct this sentence. (See the Line: 273-300).

33. l. 237: this is very interesting and quite surprising. Can you reference other works in which the diurnal cycle of ERA5 cloud cover has also been found so weak?
**Response:** Thank you for pointing out this issue, we check the original figures and datasets, and correct some bugs in the result (see the Fig. 2). In the revised manuscript, the very weak diurnal cycle of total cloud cover from ERA5 than those from other datasets is corrected, and the daily range of total cloud cover from the ERA5 exceeds 0.07 over all regions. Over the southwestern TP, the diurnal cycle of total

cloud cover from the ERA5 is strongest and the amplitude even reach 0.19. On average, the daily amplitude of the total cloud cover from ERA5 is about 0.13 over whole TP. It is similar with the result from Lei et al. (2020), who found the daily range of total cloud cover from the ERA5 over TP region is around 0.15 by using one month's data from ERA5. Overall, diurnal cycle of total cloud cover from ERA5 is still weaker compared with those from Himawari-8, ISCCP and CATS, but is comparable with those of MERRA-2 and MEM. Here, we are also very sorry to mislead reviewer due to some bugs in the data processing. In the revised manuscript, we correct some errors and revise the related discussion. (See the Line: 273-300).

34. l. 240: do you mean "half as large"? As I see it the Himawari cloud cover is much smaller than the CATS cloud cover.
**Response:** Here, our mean is that the amplitude of diurnal cycle of total cloud cover from Himawari-8 is nearly one and a half times as large as that of CATS. We revise the related sentence to make it clearer. (See the Line: 232).

35. l. 241: the sentence mentions western TP (figures 2a and 2c), and references figure 2b???
**Response:** We are sorry to mislead the reviewer. Indeed, the logic of discussion about this figure is disordered in the previous manuscript. In the revised manuscript, we reconstruct this sentence. (See the Line:273-300).

36. l. 244: why "partly"? What other explanations are there? If all instruments had the same detection sensitivity and coverage, wouldn't their cloud covers match exactly?
**Response:** Thanks for your comment! As stated by reviewer in the comment # 68, we can't completely reconcile cloud covers observed with instruments based on different observation methods, thus difference always exists between instruments or observation methods. But, detection limitations of different sensors still possibly contribute part of difference of total cloud cover. Such as, passive sensor is hard to detect the optically thin cloud, but lidar can, especially during the night-time.

37. l. 257-259: Since H8 features the lowest cloud cover of all datasets here, and that ERA5 and ISCCP cloud covers match quite well CATS cloud cover, wouldn't it be more appropriate to say that H8 underestimates the cloud cover by 10% compared to ERA5 and 20% compared to compared to ISCCP?
**Response:** We agree with reviewer. In the study of Lei et al. (2020), they consider the H8 as the "truth", thus they conclude that ERA5 and ISCCP overestimate 10% and 20% of the total cloud cover over TP compared to the Himawari-8, respectively. Here, we only cite their statement. Indeed, our results indicate that ERA5 and ISCCP cloud covers match quite well CATS cloud cover than that from H8. It thus refers to we consider which dataset as "truth". To make the description clearer, we add one sentence in the revised manuscript: "However, our results indicate that the ERA5 and ISCCP have more closer cloud covers with those from CATS compared with that of Himawari-8. It means that Himawari-8 should underestimate the total cloud cover than ERA5 and ISCCP." (See the Line: 344-348).

38. l. 261-286: it's not clear to me what is gained by this analysis. It is not referred to at all in the rest of the paper. Please consider what would be effectively lost by moving this paragraph to an appendix?
**Response:** Thanks for your comment! In the previous version, we use the Taylor diagram in order to discuss the spatial consistency of total cloud cover from passive satellites, reanalyses and models with

CATS observations at different local times. Indeed, it is not referred to at all in the rest of the paper. Based on the comment from reviewer, we move this paragraph to the appendix part.

39. Section 3.2: this part is quite long and could benefit from being split up

**Response:** Thank you for your comment!Based on the suggestion from reviewer, we split up this section in two parts. That is, section 3.2: Comparison of cloud vertical distribution from different datasets; Section 3.3: Diurnal cycle of cloud vertical distribution.

40. l. 305: the differences appear smaller in Figure 4a, and bigger in the subregions. This suggests that the subregions are perhaps too small for the sampling of CATS observations to be representative of what is going on with the clouds in that region. Also, you don't specify over which period you've used the CALIPSO/CloudSat and CALIPSO dataset. If you've used anything longer than 2015-2017, differences with CATS could come from that too.

**Response:** Thank you for your comment!In our revised manuscript, we correct some bugs in some results and figures, it results in a little bit change of pattern of cloud vertical profile from these datasets than those of previous version. The corrects include: (1) Removing the profiles that are fully attenuated below opaque layers for CATS and CALIPSO; (2) 2B-GEOPROF-LIDAR product is only released till the end of 2016 at the time of writing, but now we can extend the observation period of 2B-GEOPROF-LIDAR dataset and CALIPSO observation to the Oct./2017, and keep consistency with that of CATS. (3) Correcting the bug in pressure-height converting for MERRA2 and ERA5. (4) In the previous version, the CATS L2O 5 km profile product, which is used to provide the tropopause height information (see the Comment #47), have some loss at observation period due to link issue, but now we already fill the miss data. After these corrections, we replot the Fig.4 and find that CALIPSO still agrees well with CATS about the cloud vertical distribution, especially below the peak height over the northwestern and southwestern parts of the TP (Fig. 4b and Fig. 4d). But, the small negative difference of cloud vertical distribution between CATS and CALIPSO is almost consistent over four subregions, and possibly comes from the spatio-temporal matching process, mostly. We also provide the cloudy and total sample number profiles of the hour closest to the CloudSat and CALIPSO daytime overpass time over every subregion during the entire time periods of CATS (that is, from March/2015 to Oct./2017). Fig.R4 indicates that the total samples above 6km exceed 3000over four subregions, and corresponding cloudy samples also exceed 500 above 6km except southwestern part of TP. This suggests that statistical samples are enough over most of regions.

[Figure]

**Figure R4: (a)The sample size before (blue lines) and after (red lines) removing the profiles that are fully attenuated below opaque layers and the difference between them (yellow lines) at the hour closest to the CloudSat and CALIPSO daytime overpass time for CATS. (b) The sample size of cloudy samples.**

41. l. 310-314: As you say, this bias is probably due to optically thick clouds masking the bottom of the atmosphere in CALIPSO data, but its impact should be limited by taking it into account when documenting the number of available profiles in every height level, as suggested in my main comment.

**Response:** Thank you for your comment again! Following the suggestion from reviewer, the calculation of cloud cover for CALIPSO at a given height bin is same as that of CATS, that is, removing the profiles that are fully attenuated below opaque layers from the total number of profiles (see the Fig.R5). In addition, we also extend the observation period of 2B-GEOPROF-LIDAR dataset and CALIPSO observation to the Oct./2017, and keep consistency with that of CATS. As shown in the Fig.R5 and R1, the underestimation of low-level clouds by CATS and CALIPSO due to optical extinction from higher clouds can be slightly improved via removing those profiles that are fully attenuated below opaque layers from the total number of profiles. Compared with 2B-GEOPROF-lidar, however, we find that CATS and CALIPSO datasets still obviously underestimate the cloud cover at middle and low atmosphere levels, and the bias of cloud cover even reaches 0.2 and 0.15 at 8 km and 4 km over the southeastern TP (see the Fig. 4e in the revised manuscript), respectively.

[Figure]

**Figure R5: The same as Fig. R4 but for CALIPSO.**

[Figure]

**Figure R1: The difference between revised cloud vertical distribution and the original results from CALIPSO (blue lines) and CATS (red lines) at the hour closest to the CloudSat and CALIPSO daytime overpass time. (a) The whole TP (b) The northwestern TP (c) The northeastern TP (d) The southwestern TP (e) The southeastern TP. The regions are divided by latitude and longitude lines of 33°N and 89°E (shown in Fig. 1).**

42. l. 317: "status"

**Response:** It is corrected to "stratus" in the revised manuscript. (See the Line: 387).

43. l. 320: why don't you discuss the vertical distribution predicted by CMIP6 models? At least acknowledge why you think it is not a good idea

**Response:** Thanks for your comment! In fact, we plan to discuss the vertical distribution predicted by CMIP6 models in the previous manuscript. However, we found that only 4 CMIP6 models can provide the cloud vertical distribution with 3-hourly temporal resolution, meantime, two of them (MRI-ESM2-0, IPSL-CM6A-LR) provide only one year of data (e.g., 2008). The above is talking about the CMIP6 historical runs. For the RCP8.5 runs, there is no model can provide cloud cover profile with 3-hourly temporal resolution. In fact, it is interesting to compare the cloud vertical distribution predicted by CMIP6 models (especially with lidar simulator) with those from CATS or CALIPSO/CloudSat. However, current study still can't perform this analysis due to cloud vertical distribution is unavailable in CMIP6 models with 3-hourly temporal resolution.

44. l. 336-337: this is an important result I think.

**Response:** Thanks for your comment! In fact, some studies have indicated that the vertical profile of cloud cover in the MERRA-2 is also obviously underestimated over other regions (e.g., Miao et al., 2019). Miao et al. (2019) found that MERRA-2 showed better performance for high-level clouds but underestimated low- and mid-level clouds compared with CALIPSO/CloudSat (Please see manuscript Line: 409-411). They found the biases occur in reanalyses touches the basis of cloudiness parameterization in general circulation models. For most current models, the cloud cover of each layer is diagnosed by either an empirical formula based on relative humidity or a statistical scheme based on probability density functions. The key to parameterization of cloud cover depends on how to properly consider the sub-grid scale variation of humidity. With the comparison of "critical relative humidity" from MERRA-2 and CALIPSO/CloudSat, they suggested that poor specification or parameterization of critical relative humidity is responsible for the biases.

45. l. 353, l. 363: we know that in your results CALIPSO understimates low-level clouds due to optical extinction from higher clouds, the same is probably happening for CATS data here. This effect might get less important if data analysis is revised (see main comment #1)

**Response:** Thank you for your comment again! The detailed response please see the comment #1, and #41. We also revise the related figures, results and their discussions in the revised manuscript.

46. l. 366: Do you have confidence in these results? Could you check its robustness by eg plotting out the number of profiles that are sampled by CATS over that region in that time period? Does it appear in all seasons? If you find it is robust, could you propose a mechanism responsible for producing this weird-looking sudden +7km increase in cloud altitude at 6PM LT (and its subsequent rapid decrease)?

**Response:**

We very thank reviewer for providing detailed comments and suggests. After removing the profiles that are fully attenuated below opaque layers and correcting some bugs in our data processing, we replot the Fig.5 and find the weird-looking large cloud cover at 18:00 LT (12:00 UTC) around 14 km become smaller. Meantime, CATS observes a high cloud cover over the southwestern TP between 11-14 km at approximately 13:00 UTC. We plot the total and cloudy sample number profiles at each hour of each subregion (see Fig.R8), and find the total sample numbers around 12:00-13:00 UTC are obvious less than those of other hours over the southwestern part of TP. By checking the cloudy samples around 14 km, we also find that wide anvil clouds contribute to the large high-level cloud cover. In summary, due to the limited total sample, this result is not as robust as other times. In the revised manuscript, we add some discussion about this issue. (See the Line:444-446).

[Figure]

**Figure R6: The hourly vertical distribution of cloudy sample number and total sample number of CATS. The red lines represent the tropopause height. The first to fourth lines represent the results over the northwestern TP, the northeastern TP, the southwestern TP and the southeastern TP, respectively. The regions are divided by latitude and longitude lines of 33°N and 89°E (shown in Fig. 1).**

47. l. 374: How did you obtain the diurnal cycle of tropopause height that is described here? What is the original data source? How was it processed?

**Response:** Thank you for your comment! The tropopause height is obtained from CATS level 2 operational (L2O) 5km profile products. The original data is provided by MERRA-2 reanalysis data, which is interpolated to the CATS 5 km L2O horizontal resolution (see CATS L2O Profile Products Quality Statements: Version 3.00, available online at https://cats.gsfc.nasa.gov/media/docs/CATS_QS_L2O_Profile_3.00.pdf). Similar with total cloud cover,

we gather the tropopause height information from all profiles in each subregion and calculated the hourly average over the entire observation period of CATS. These information is added in the revised manuscript (see 453-459).

48. l. 388: "TAU" might look better as a greek letter.
**Response:** It is corrected to "$\tau$" in the revised manuscript. (See the Line: 473-474).

49. l. 433: do you mean that overshooting over the TP can increase polar ozone consumption? Could you expand on that by explaining the mechanism?
**Response:** Thank you for your comment! This problem involves the complex dynamic and chemical processes. Overshooting is an important part of stratospheric-tropospheric exchange processes over the TP (Tian et al., 2011). The transport of chemical tracers have contributed to the increasing of stratospheric water vapor (Oltmans and Hofmann, 1995). Both oxidation of stratospheric methane and direct transport of water vapor from the troposphere contribute to the increase in stratospheric water vapor. And, this part of the stratospheric atmosphere is then transported to high latitudes by large-scale meridional circulation (e.g., Brewer-Dobson circulation) (Butchart, 2014). In the polar regions, the stratospheric water vapor concentration determines the critical temperature below which heterogeneous reactions on cold aerosols become important (the mechanism driving enhanced ozone depletion) and the temperature of the Arctic vortex itself. The above is one possible mechanism that overshooting over the TP can increase polar ozone consumption. We add a brief description about this mechanism in the revised manuscript: "Overshooting clouds driven by convection activities can affect the material exchange between tropospheric and stratospheric signals (Tian et al., 2011). Both water vapor and oxidation of stratospheric methane directly transported from the troposphere contribute to the increase in stratospheric water vapor. On the one hand, increasing stratospheric water vapor exacerbates the greenhouse effect (Forster and Shine, 2002). On the other hand, stratospheric water vapor can be transported to high latitudes by large-scale meridional circulation (e.g., Brewer-Dobson circulation) (Butchart, 2014). In the polar regions, the stratospheric water vapor concentration determines the critical temperature below which heterogeneous reactions on cold aerosols become important (the mechanism driving enhanced ozone depletion) and the temperature of the Arctic vortex itself, thus increasing stratospheric water vapor also enhances polar ozone consumption (Kirk-Davidoff et al., 1999; Luo et al., 2011)." (See the Line: 519-527).

50. l. 435: why should your results be considered preliminary? What is it that you don't trust here?
**Response:** We are sorry that our expression has caused confusion to the reviewer. In this study, we only explore the diurnal variations of overshooting clouds over the TP, but the impacts of diurnal variations of overshooting clouds on radiative budget and stratospheric-tropospheric exchange processes have not been discussed. Thus, we consider this is a preliminarily analysis. About the diurnal cycle of cloud cover above the tropopause, we use the similar method to identify the overshooting clouds with that from Dauhut et al. (2020). More detailed can be found in the Response 51. In the revised manuscript, we also delete the word "preliminary" to avoid ambiguity.

51. l. 437: As I'm sure you know, this kind of analysis is strongly dependent on the dataset considered for the tropopause altitude. It makes it even more problematic that you do not explain how this tropopause altitude was obtained and how comparisons with cloud altitudes were performed. Do you compare cloud

covers and tropopause altitudes as local-hour averages over the entire period of CATS operation? Or do you perform overshooting detection on individual profiles (as Dauhut et al. 2020 did)?

**Response:** We agree with the reviewer. Different results of overshooting clouds can be obtained by using different tropopause altitude datasets. This can be shown by the different frequency of convective overshooting calculated from COSMIC, ERA5, JRA-55 and MERRA-2 data (Sun et al., 2021). We apologize for the lack of description of the data source of tropopause height, so as to cause confusion to readers. The tropopause height is obtained from CATS L2O profile data. The original data is provided by MERRA-2 reanalysis data, which is interpolated to the CATS 5 km L2O horizontal resolution (see CATS L2O Profile Products Quality Statements: Version 3.00, available online at https://cats.gsfc.nasa.gov/media/docs/CATS_QS_L2O_Profile_3.00.pdf). Similar with total cloud cover, we gather the tropopause height information from all profiles in each subregion and calculated the hourly average over the entire observation period of CATS. The related description about tropopause height data is add in revised manuscript Section 3.3. (See the Line: 453-459).

To calculate the overshooting cloud cover, we perform overshooting detection on individual profiles. For each profile, we use 'Feature Type Score' to determine the layers of the clouds. And we use the tropopause height provided by CATS L2O profile data of the same profile to judge that if there is a cloudy layer top height is over tropopause height. If the cloudy layer top is higher than tropopause, next step we need to check this layer base height. As the cloud-aerosol discrimination algorithm cannot be applied to the CATS L2O layer entirely above the tropopause (like in CALIPSO). we consider only clouds with a base in the tropopause and a top in the stratosphere as overshooting clouds. The method above is the same as Dauhut et al. (2020) except we use different tropopause height data source. Dauhut et al. (2020) used ERA5 temperature and pressure profiles to computed the vertical lapse rate profile and then obtained the tropopause height. The differences between the tropopause height from MERRA-2 and ERA5 is within 0.6 km over the TP (Sun et al., 2021). Shown in the comparison of the tropopause calculated by COSMIC observation data, and MERRA-2, ERA5 (Sun et al., 2021), the spatial distribution of COSMIC and ERA5 are similar, and the tropopause MERRA-2 is a little higher than COSMIC. The tropopause height biases may have some impact on our results for the diurnal variations of overshooting clouds. We revise the description in revised manuscript Line: 531-534: "By following the methods of Dauhut et al. (2020), we perform overshooting detection on individual profiles. Because the cloud-aerosol discrimination algorithm cannot be applied to the CATS L2O layer entirely above the tropopause (Pan and Munchak, 2011), we only consider the cloud with base lower than tropopause height and top higher than tropopause height as overshooting cloud as did by Dauhut et al. (2020)."

52. Section 3.4: I find it problematic that the diurnal cycle of cloud cover of cirrus clouds (at 10km above the surface) is compared to the diurnal cycle of surface properties (T2m and 10m wind speed) as if the latter were driving the former. Please clarify the description as to explain that cirrus cloud cover and surface properties might all be driven by the same underlying mechanism.

**Response:** Thank you for your comment! We are sorry for the confused expression. It is corrected to: "In this section, we further analyse the correlation of the diurnal cycle between the total cloud cover (and cirrus cover) from CATS dataset and related meteorological factors in the ERA5 datasets over the TP." (See the Line: 562-565). The correlation coefficients between the diurnal cycle of total cloud cover and 2-m temperature, 10-m wind speed, and vertically integrated divergence of moisture flux are analysed in the first paragraph of Section 3.4 shown in Fig. 8. And the correlation coefficients between the diurnal cycle of cirrus and 250 hPa relative humidity, 2-m temperature, and 250 hPa vertical velocity are

analysed in the second paragraph shown in Fig. 9. Due to bugs in the calculation of diurnal variations of cirrus and meteorological factors, the results are corrected in revised manuscript.

We agree with reviewer that the diurnal variations of all the meteorological factors here and clouds are under the influence of the diurnal variations of solar radiation. But it needs to be made clear that 2-m temperature and cirrus diurnal variations are not completely unrelated, and do not just coincidentally share similar diurnal variations in response to solar radiation. The following mechanisms can explain the relationship between them. A previous work indicated that surface air temperature is likely to promote the formation of cirrus through at least two effects (Kent et al., 1995). On the one hand, the equilibrium water vapor mixing ratio increases with temperature based on the Clausius-Clapeyron equation, which contribute to the increase of ice water content directly. On the other hand, the rise of temperature increases convective available potential energy (CAPE), which is required in the transport of ice particles to the upper troposphere to form cirrus. (even in diurnal timescales) (Williams and Renno, 1993). Shown in Fig. 9, the promotion effect of high 2-m temperature on cirrus is more reflected in opaque cirrus. Theseopaque cirrus are always formed by the deep convection outflow (He et al., 2013) and have similar diurnal variations with deep convection (Devasthale and Fueglistaler, 2010). On the contrary, subvisible cirrus are negatively correlated with 2-m temperature, which attributed to the detrainment from deep convection and evolution cost time. The above mechanisms between 2-m temperature and cirrus might only explain part of cirrus related to convection. We clarify the the description of results in Fig. 9 in revised manuscript. (See the Line:649-654).

53. l. 452: "standardized": What does this mean? How did you get the standardized cloud column?
**Response:** Thanks for your comment! "standardized" means all factors including cloud cover and meteorological factors shown in Fig. 8 are standardized by z-score transformation.
The following is a description of z-score:
For sample data with mean $\overline{X}$ and standard deviation $S$, the z-score of a data point $x$ is

$$z = \frac{(x - \overline{X})}{S}$$

z-scores measure the distance of a data point from the mean in terms of the standard deviation. This is also called standardization of data. The standardized data set has mean 0 and standard deviation 1, and retains the shape properties of the original data set (same skewness and kurtosis). We can use z-scores to put data on the same scale before further analysis. This lets us to compare two or more data sets with different units. The following expression about this method is added in revised manuscript: "All factors, including total cloud cover and meteorological factors, are standardized using z-score transformation for comparison. Z-scores measure the distance of a data point from the mean in terms of the standard deviation. This method is used for the comparison of datasets with different units and retains the shape properties of the original datasets (same skewness and kurtosis)." (See the Line: 565-569).

54. l. 454: "statistical results": which statistical results? If you're referring to the correlation coefficients you are about to describe, please move that statement after their description
**Response:** Thanks for your comment! Based on your suggestion, we move the statement after their description. (See the Line: 577-578).

55. l. 470: "the correlation provides only limited insights": so, what are they good for?
**Response:** Here, our understanding is that the correlation analysis can be used to quantify the degree of

correlation between two variables, but the causal relationship between them is hard to build. This is because two variables without any relationship still can exhibit high correlation due to they are both affected by same factor.

56. l. 483: "radiational" radiative
**Response:** It is corrected in the revised manuscript. (See the Line: 600).

57. l. 484: 250hPa and 2m are quite different altitude levels. Please see my comment for Section 3.4 above
**Response:** Thank you very much for your careful consideration. We fully understand the reviewer's concern. The high-level cirrus seem to be more easily associated with meteorological factors at 250hPa than with surface properties. However, the generation of cirrus is closely related to the convective activity and high-altitude ice production which promoted by surface heating. More detailed mechanisms can be found in Response 52. Meanwhile, the formation mechanisms of cirrus clouds differ according to different types (Heymsfield et al., 2017). From our results (Fig. 9d), only the opaque cirrus has a positive correlation with the diurnal variations of 2-m temperature. Sassen et al. (2003) also indicates that except for the cirrus freshly generated from thunderstorm anvils linked to diurnal cycles, the cirrus clouds of the upper troposphere that are normally decoupled from local surface heating effects. Overall, there are mechanisms that account for the effect of 2-m temperature on diurnal variation of cirrus, but only for a subset of cirrus associated with convection.

58. l. 488: midnight and 03:00LT are different things
**Response:** It is removed in the revised manuscript.

59. l. 503-504: I think there is a misunderstanding here. You write that changes in the temperature at 2m somehow drives the diurnal evolution of cirrus cloud cover. I find this hard to believe. How do you propose that would work? Would surface infrared emission somehow lead to changes in cirrus cloud cover? What i could believe is, that the diurnal evolution of the temperature at 2m and of cirrus cloud cover are both driven by the same mechanism, which is heating from solar illumination. If you had access to the temperature at 250hPa, that might be easier to show. This is unfortunately harder to get. Please check your explanations.
**Response:** Thanks for your comments! The 2-m temperature can influence the diurnal variation of cirrus cloud cover by affecting convective activity and high-altitude ice production. The detailed mechanisms can be found in Response 52 and Response 57. And the diurnal variation of opaque cirrus is associated with deep convection, which is promoted by high surface air temperature. As ground-based lidar measurements over the TP show that cirrus with optical thickness above 0.3 are always observed near deep convection (He et al., 2013). And the diurnal cycle of deep convection over the TP obtained by Meteosat-5 data (Devasthale and Fueglistaler, 2010) is similar with our results of opaque cirrus with a peak around 10:00-12:00 UTC. Based on the above two points, we infer that the changes in the temperature at 2m can influence deep convection and then influence the formation of opaque cirrus.
In addition, as the reviewer mentioned, cirrus cloud formation and development are in part influenced by radiative effects. The surface infrared emission can warm the lower portions of a cirrus cloud and consequently produce convection and turbulence of sufficient strength to maintain or enhance the cirrus (Heymsfield et al., 2017). From the point of view of radiative effects, Ackerman et al. (1988) find that

solar heating in anvils is shown to be less important than infrared heating but net negligible, especially for cirrus with large IWCs. For these clouds, both the heating rate profile and the total solar heating vary substantially with solar zenith angle. We strongly agree with the reviewer's opinion that the most fundamental driving force of diurnal variation is solar radiation. Thus, we add following description: "Although there is a correlation between cloud cover of cirrus and meteorological factors, in fact, the diurnal variations of clouds and these meteorological factors are both influenced by the diurnal variations of solar radiation." to the revised manuscript Line: 611-613.

60. l. 514 this "air mass uplift" is what happens in deep convection. Please clarify your text here.
**Response:** Thanks for your comment! This confused description is removed in the revised manuscript.

61. l. 515 "positive correlation" is technically true but could be deceptive. A 0.01 coefficient correlation is a positive correlation, but it is not high enough to be meaningful. Please revise.
**Response:** We agree with reviewer. In the revised manuscript, we revise the related sentence as: " The diurnal variation of total cirrus cloud cover is only significantly positive correlation with 250 hPa relative humidity at a 90% confidence level (correlation coefficient is 0.77, see Fig. 9a)". (See the Line: 605-606).

62. l. 532: Compared to CATS they underestimate cloud cover almost as much as H8. Please mention that H8 underestimates the cloud cover as well.
**Response:** Thanks for your comment! Here, the second point of conclusions mainly describes the vertical distribution of clouds. Based on suggestion from reviewer, we add the related description in the first point of conclusions. (See the Line: 698).

63. l. 540: peaks
**Response:** The "peak" is corrected to "peaks" in revised manuscript.

64. l. 543: "Over 7% of the subvisible cirrus clouds exist at night": Does this mean that 93% of subvisible cirrus are found during daytime? Or do you actually mean that the cloud cover of subvisible cirrus reaches 7% at night? Is that a lot or a little compared to the daily average? Please add some details to help the reader who will only read the conclusion
**Response:** We are sorry to mislead the reviewer. Yes, the 7% indicates the cloud cover of subvisible cirrus. In the revised manuscript, we add one sentence: "In particular, the cloud cover of subvisible cirrus is approximately 0.07 at night (15:00-23:00 UTC), twice as large as during daytime." (See the Line: 712-713).

65. l. 543: what is difficult to detect?
**Response:** We are sorry to mislead the reviewer. This sentence is revised as: "But, these subvisible cirrus clouds are still difficult to be detected during nighttime by using passive methods". (See the Line: 713-714).

66. l. 555-556: this is an important result
**Response:** Thanks for your comment!

67. l. 556-557: this supposes that (global-scale) climate change is strongly dependent on the cloud diurnal cycle in the TP region. I'm not sure this has been conclusively demonstrated

**Response:** We agree with reviewer. Indeed, this statement has not been conclusively demonstrated. Here, we just want to indicate that there is large difference in the diurnal cloud cycle between these datasets, and the impact of diurnal variation of cloud cover on radiative budget should be considered in models. In the revised manuscript, however, we delete this sentence in order to avoid an arbitrary conclusion.

68. l. 558-560: in a general sense, it is unrealistic to assume the cloud cover can be defined outside of actual instruments with their own detection sensitivities. The cloud cover does not exist without an instrument to measure it. It will always be impossible to completely reconcile cloud covers observed with instruments based on different observation methods.

**Response:** We agree with reviewer. We can't completely reconcile cloud covers observed with instruments based on different observation methods, thus difference always exists between instruments or observation methods. Thus, we revise the related statement: "Of course, it is impossible to completely reconcile cloud covers observed with instruments based on different observation methods. However, the part of total cloud cover difference between different datasets is possibly caused by following problems:" (See the Line: 732-734).

69. l. 565: detection based on solar backscatter will still be daytime-only, and subvisible cirrus are more frequent over nighttime, as you showed in your results, so it doesn't solve the problem

**Response:** We agree with reviewer. Indeed, subvisible cirrus are more frequent over nighttime of the TP region. The detection of subvisible cirrus based on the backscattered solar radiation is only feasible during the daytime. Until now, the space-based lidar possible is the most effective tool in detecting the optically thin cloud during the nighttime. In the revised manuscript, we add one sentence: ", but this new approach is also only available during the daytime. Over the TP region, our results indicate the subvisible cirrus clouds are more frequent during nighttime. It means that the detection of subvisible cirrus based on the backscattered solar radiation still cannot reduce the uncertainty of observation during nighttime" (See Line 743-745).

70. l. 569: what is GRAPES-GFS and how is it relevant to the results you present here? Please avoid introducing unrelated elements right before the conclusion

**Response:** We agree with reviewer. Here, we just want to give an example of the advantage of the physical processes of cloud formation in cloud cover simulations. It is really irrelevant to the results from our study. Based on your suggestion, we delete this unrelated element in the revised manuscript.

71. l. 584: how come Gasparini 2019, Zou 2021 and Zhang 2021 were able to propose mechanisms responsible for the formation of cirrus clouds, and you're not? I don't think they were equipped with more data than you are. Here you are using non-sunsynchronous spaceborne lidar data from CATS and CALIOP, output from climate models, ISCCP and geostationary imagery, and not one but two reanalyses datasets. I'd say your dataset is pretty comprehensive. You have all the elements to propose an interpretation of the processes responsible for cirrus creation.

**Response:** We very thank the suggestion from reviewer. Gasparini et al. (2019) use cloud-resolving model to explore anvil cirrus evolution. Zou et al. (2021) analyse the spatial and temporal relation between cirrus and deep convection, cirrus and gravity waves based on CALIPSO and AIRS. The high

similarity between them suggests that they are mechanistically linked. Similarly, Zhang et al. (2020) work out the mechanism by analyzing the distribution characteristics of cirrus and meteorological factors based on CALIPSO and reanalysis. Based on the suggestion, we will try to combine these datasets to discuss the formation processes of cirrus clouds in the future work.

72. l. 585: "Further comprehensive investigations...": again, I don't think investigations can get much more comprehensive than yours.
**Response:** Thanks for your comment. We already delete this sentence.

73. l. 587: are you suggesting that aerosol loading could be one of the major influences driving the diurnal cycle of cirrus clouds when averaged over many years?
**Response:** In our study, we only discuss the correlation of diurnal cycle between cirrus and meteorological factors. However, some previous studies have indicated that diurnal cycles of cloud properties (e.g., cloud droplet size and cloud liquid water path) is related with the variation of aerosol loading in their study periods (Matsui et al., 2006; Ntwali and Chen, 2018), but these studies don't address the impact of meteorological factors on the diurnal cycle of cirrus clouds. By using the 33 months of dust aerosols extinction coefficient and meteorological factors, wang et al. (2022) s how a robust dependence of diurnal cycle of supercooled water cloud cover on the variation of dust aerosol extinction coefficient instead of other dust load indicators and meteorological parameter. These results demonstrate that the aerosol loading can affect the diurnal cycle of cloud cover, however, whether aerosol loading over the TP region is the major driving factor of diurnal cycle of cirrus clouds is still unclear. Thus, future work also should pay more attentions on the impact of aerosol on the diurnal cycle of cirrus clouds over the TP region. Related discussions are added in the revised manuscript (see Line:765-773).

74. l. 591-596: all the instances of "is" here should be replaced by "are" (data is plural)
**Response:** They are corrected in revised manuscript.

75. l. 602: "and carried them out". Carried what out?
**Response:** Thanks for your comment! It is corrected as: "YZ and JL organized the paper and performed related analysis". (See the Line: 809).

76. l. 603: "maintain"?
**Response:** Thanks for your comment! It is corrected in revised manuscript. (See the Line: 811).

77. Check the order of references. For instance, the many Li et al. references are not in chronological order. They are not alone with this problem.
**Response:** Thank you for pointing out this issue. The order of references is corrected in revised manuscript.

78. Figure 1: when first looking at Figure 1, I would have liked to see a figure showing maps of correlation coefficients between each pair of datasets -- CATS vs ISSCP, CATS vs H8, CATS vs ERA5, etc. As a grid. It would help quickly visualise in which regions the diurnal cycle of which datasets are well/not well correlated. Please consider building this figure and including it if it brings anything of value to the discussion.

**Response:** We very thank reviewer for providing this important suggestion. Based on your suggestion, we add the correlation of diurnal cycle of total cloud cover between these datasets (see Fig. R7). Meantime, related discussion is also added in the revised manuscript (see the Line: 301-319): "To find out in which regions the diurnal cycle of which datasets are well correlated with CATS, Figure 3 further shows the spatial distribution of correlation coefficients of diurnal cycle for total cloud cover between CATS and other datasets in a $2 \times 2°$ grid box. As shown in the Fig.3, ISCCP exhibits best correlation with CATS, and the correlation coefficient (at 90% confidence level) is even greater than 0.5 over the most areas (Fig.3a), especially over the central part of TP. The diurnal cycle of total cloud cover from the Himawari-8 obviously positive correlates with that of CATS over the most part of TP, but the correlation is almost insignificant over TP region (Fig. 3b). It may be caused partly by the limited observation hours from Himawari-8. Here, it is worth noting that because the cloud cover calculation of CATS needs to ensure that there are enough profiles in each grid, it is difficult to split more sample points by months or seasons for correlation analysis. Therefore, the correlation analysis here can only be used as a reference to some extent. Similar with ISCCP, ERA5 also shows significant positive correlation with diurnal cycle of total cloud cover of CATS over the central and western parts of TP (see Fig. 3c), but we also find the ERA5 is the only dataset that exhibits opposite diurnal variation with CATS over the eastern part of TP, and correlation coefficient (at 90% confidence level) even reaches -0.9. As stated in Fig.2, MERRA-2 and MEM show almost synchronous diurnal variations of total cloud cover, resulting in the correlations coefficients of diurnal cycle from them with CATS are very similar, that is, there is a significant positive correlation coefficient over the northern part of TP (Fig. 3d and 3e). Although Fig.3 indicates that ISCCP exhibits closer diurnal cycle of total cloud cover with that of CATS over most part of TP, the averaged spatial consistency of total cloud cover at all times between ISCCP and CATS is lowest compared with those from ERA5, Himawari-8, MERRA-2 and MEM (see Fig. A1 in the appendix). In summary, above statistical results show that total cloud cover from multiple sources exhibits considerable regional differences in the phase and magnitude of the diurnal cycle."

[Figure]

**Figure R7: The spatial distribution of correlation coefficients of diurnal cycle for total cloud cover between CATS and other datasets. The grids are marked with "+" if the correlation at these grids pass the significance test by 90%. Only cloud cover results at 00:00 UTC, 03:00 UTC, 06:00 UTC, 09:00 UTC are used in the correlation between CATS and Himawari-8, as only daytime cloud cover is available from Himawari-8.**

79. Figure 4: It could be useful for the reader if you could locate for each region the topmost surface height altitude. I expect the TP surface altitude above the sea level to be quite high, but its variation across the TP is unknown to me. I'm assuming here that all the altitudes shown in the paper are above the mean sea level and not in reference to the surface, please make that explicit somewhere in the text.

**Response:** We highly appreciate the reviewer for valuable comments. In our manuscript, all the altitudes are above the mean sea level. Based on the suggestion from reviewer, we also use the DEM elevation in CATS L2O Layer products to add the topmost and bottommost surface height altitudes of each region in the Fig.4 (see the Fig.4 in the revised manuscript). This DEM elevation is the surface elevation at each laser IFOV footprint, in kilometres above local mean sea level. The DEM is from JPL created for CloudSat and CALIPSO, and it has a horizontal resolution of ~500 m.

80. Figure 5: Where do the tropopause altitudes come from? How were they processed?

**Response:** Thank you for your comment! We already add the related information of tropopause altitudes in the revised manuscript. Detailed response, please see the comments #47 and #51.

81. Figure 6: please find a way to show the optical depths of cirrus clouds in each subplot. Please explore ways to make the y-axis limits of the 4 figures as consistent as possible.

**Response:** Thanks for your comments! We have revised the y-axis limits of Figure 6, please see the revised manuscript.

82. Figure 7: I am particularly concerned by the fact that apart from strong peaks (cover > 0.01), the small overshooting fractions appear to follow a pattern that makes them maximum at 7:00, 9:00, 11:00, 13:00, 15:00, 17:00, 19:00, 21:00... and minimum at 6:00, 8:00, 10:00... etc. Could you please check that this is not an artifact, for instance related to the variation of the number of available profiles at each time step? In a more general way, could you somehow discuss the uncertainty of these results? For instance, if at a given local time only one profile features overshooting, I'm not sure if the result could be called representative. Could you quantify the cloud cover that would be reached if only one profile was found as overshooting?

**Response:** Many thanks for the reviewer's suggestion. In the revised manuscript, we correct some bugs in the data processing, and replot the diurnal cycle of overshooting cloud cover at different subregions and seasons (see Fig. R9). Indeed, the available cloudy samples are less in different subregions and seasons. To add the robustness of statistical result, we combine the all samples in subregions and only provide the diurnal cycle of overshooting cloud cover over whole TP (see Fig. 7 in the revised manuscript).

Over the TP, the averaged cloud cover of overshooting cloud is higher at night and has a maximum value at 16:00 UTC (22:00 LT), and its value is approximate 0.013. The overshooting cloud cover over the TP is smaller than that in the tropics (Dauhut et al., 2020) with one order of magnitude. Sun et al. (2021) also found this difference in magnitude of occurrence frequency of convective overshooting between TP and tropical and subtropical areas based on TRMM. Besides the 16:00 UTC, overshooting

cloud cover also has large value around 10:00 UTC (16:00 LT), 13:00 UTC (19:00 LT), 20:00 UTC (02:00 LT) and 22:00 UTC (04:00 LT). Multiple peaks in diurnal cycle are possibly caused by the reginal difference of overshooting cloud. Such as, peak value at 16:00 UTC is linked with the overshooting cloud over the southern TP (Fig. R8a), especially during the summer (Fig. R8b). The peak value at 13:00 UTC is possibly related with the overshooting cloud over the southeastern and northwestern parts of TP (Fig. R8a), especially during the winter (Fig. R8b). Here, it is worth noting that the seasonal and regional results in Fig.R8 are not robust as those in the Fig. 7 due to fewer cloudy sample (see Fig. R9). However, even if Fig.7 reveals the diurnal cycle of overshooting cloud cover over the whole TP to a certain extent, the statistical result is still noisy due to the overshooting cloud sample number only approaches 300 at 16:00 UTC and is less than 100 most of time (see Fig. R10). In addition, the difference in the tropopause altitude from different data source also possibly induces some uncertainties in our statistical result. For example, by comparing the tropopause height from MERRA-2, ERA5 and COSMIC observation data, sun et al.(2021) pointed out that the spatial distribution of tropopause height from COSMIC and ERA5 are similar, but the tropopause from MERRA-2 is a little higher than COSMIC. Overall, the differences between the tropopause height from MERRA-2 and ERA5 is within 0.6 km over the TP (Sun et al., 2021). It means that the overestimation in tropopause height from MERRA-2 may cause a little bit underestimation of overshooting cloud cover over TP. It is the one of possible reasons why the overshooting cloud cover over the TP is smaller than that in the tropics by Dauhut et al. (2020), who used ERA5 temperature and pressure profiles to compute the tropical tropopause height. Above discussions have been added in the revised manuscript (see Line:540-559).

[Figure]

**Figure R8 (same as the Fig. S3 in the revised manuscript): The hourly cloud cover of cirrus shooting over tropopause based on CATS for different subregions (a) and different seasons (b). The regions are divided by latitude and longitude lines of 33° N and 89° E and the boundary of TP (shown in Fig. 1). The average of the whole day of each region and season is indicated in the legend. All seasons here are northern hemisphere seasons.**

[Figure]

**Figure R9(same as the Fig. S4 in the revised manuscript): The number of overshooting cloudy profiles (a, c) and total samples (b, d) in different subregions (a, b) and different seasons (c, d).**

[Figure]

**Figure R10 (same as the Fig. S5 in the revised manuscript): The number of overshooting cloudy profiles (a) and total samples (b) over the TP.**

**References**

Ackerman, T. P., Liou, K.-N., Valero, F. P. J., and Pfister, L.: Heating Rates in Tropical Anvils, J. Atmos. Sci., 45, 1606-1623, https://doi.org/10.1175/1520-0469(1988)045<1606:Hrita>2.0.Co;2, 1988.

Butchart, N.: The Brewer-Dobson circulation, Rev. Geophys., 52, 157-184, https://doi.org/https://doi.org/10.1002/2013RG000448, 2014.

Chepfer, H., Brogniez, H., and Noel, V.: Diurnal variations of cloud and relative humidity profiles across the tropics, Sci. Rep., 9, 16045, https://doi.org/10.1038/s41598-019-52437-6, 2019.

Devasthale, A. and Fueglistaler, S.: A climatological perspective of deep convection penetrating the TTL during the Indian summer monsoon from the AVHRR and MODIS instruments, Atmos. Chem. Phys., 10, 4573-4582, https://doi.org/10.5194/acp-10-4573-2010, 2010.

Feofilov, A. G. and Stubenrauch, C. J.: Diurnal variation of high-level clouds from the synergy of AIRS and IASI space-borne infrared sounders, Atmos. Chem. Phys., 19, 13957-13972, https://doi.org/10.5194/acp-19-13957-2019, 2019.

He, Q., Li, C., Ma, J., Wang, H., Shi, G., Liang, Z., Luan, Q., Geng, F., and Zhou, X.: The Properties and Formation of Cirrus Clouds over the Tibetan Plateau Based on Summertime Lidar Measurements, J. Atmos. Sci., 70, 901-915, https://doi.org/10.1175/jas-d-12-0171.1, 2013.

Heymsfield, A. J., Krämer, M., Luebke, A., Brown, P., Cziczo, D. J., Franklin, C., Lawson, P., Lohmann, U., McFarquhar, G., Ulanowski, Z., and Van Tricht, K.: Cirrus Clouds, Meteorological Monographs, 58, 2.1-2.26, https://doi.org/10.1175/amsmonographs-d-16-0010.1, 2017.

Kent, G. S., Williams, E. R., Wang, P. H., McCormick, M. P., & Skeens, K. M.: Surface temperature related variations in tropical cirrus cloud as measured by SAGE II. J. Clim., 8(11), 2577-2594, 1995.

Kirk-Davidoff, D. B., Hintsa, E. J., Anderson, J. G., and Keith, D. W.: The effect of climate change on ozone depletion through changes in stratospheric water vapour, Nature, 402, 399-401, 1999.

Lei, Y., Letu, H., Shang, H., and Shi, J.: Cloud cover over the Tibetan Plateau and eastern China: a comparison of ERA5 and ERA-Interim with satellite observations, Clim. Dyn., 54, 2941-2957, https://doi.org/10.1007/s00382-020-05149-x, 2020.

Matsui, T., Masunaga, H., Kreidenweis, S. M., Pielke Sr., R. A., Tao, W.-K., Chin, M., and Kaufman, Y. J.: Satellite-based assessment of marine low cloud variability associated with aerosol, atmospheric stability, and the diurnal cycle, J. Geophys. Res.-Atmos., 111, https://doi.org/10.1029/2005JD006097, 2006.

Ntwali, D. and Chen, H.: Diurnal spatial distributions of aerosol optical and cloud micro-macrophysics properties in Africa based on MODIS observations, Atmos. Environ., 182, 252-262, https://doi.org/10.1016/j.atmosenv.2018.03.054, 2018.

Oltmans, S. and Hofmann, D.: Increase in lower-stratospheric water vapour at a mid-latitude Northern Hemisphere site from 1981 to 1994, Nature, 374, 146-149, 1995.

Sassen, K., Starr, D. O. C., and Uttal, T.: Mesoscale and microscale structure of cirrus clouds: Three case studies, J. Atmos. Sci., 46, 371-396, https://doi.org/10.1175/1520-0469(1989)046<0371:Mamsoc>2.0.Co;2,, 1989.

Sassen, K., Liou, K.-N., Takano, Y., and Khvorostyanov, V. I.: Diurnal effects in the composition of cirrus clouds, Geophys. Res. Lett., 30, 1539, doi:10.1029/2003GL017034, 10, 2003.

Sassen, K., Wang, Z., and Liu, D.: Cirrus clouds and deep convection in the tropics: Insights from CALIPSO and CloudSat, J. Geophys. Res.-Atmos., 114, https://doi.org/10.1029/2009JD011916, 2009.

Shang, H., Letu, H., Nakajima, T. Y., Wang, Z., Ma, R., Wang, T., Lei, Y., Ji, D., Li, S., and Shi, J.: Diurnal cycle and seasonal variation of cloud cover over the Tibetan Plateau as determined from Himawari-8 new-generation geostationary satellite data, Sci. Rep., 8, 1105, https://doi.org/10.1038/s41598-018-19431-w, 2018.

Sun, N., Fu, Y., Zhong, L., Zhao, C., and Li, R.: The Impact of Convective Overshooting on the Thermal Structure over the Tibetan Plateau in Summer Based on TRMM, COSMIC, Radiosonde, and Reanalysis Data, J. Clim., 34, 8047-8063, https://doi.org/10.1175/jcli-d-20-0849.1, 2021.

Sun, W., Videen, G., Kato, S., Lin, B., Lukashin, C., and Hu, Y.: A study of subvisual clouds and their radiation effect with a synergy of CERES, MODIS, CALIPSO, and AIRS data, J. Geophys. Res.-Atmos., 116, https://doi.org/10.1029/2011JD016422, 2011.

Tseng, H. H. and Fu, Q.: Tropical tropopause layer cirrus and its relation to tropopause, J. Quant.

Spectrosc. Radiat. Transf., 188, 118-131, https://doi.org/https://doi.org/10.1016/j.jqsrt.2016.05.029, 2017.

Wang, Y., Li, J., Zhao, Y., Li, Y., Zhao, Y., and Wu, X.: Distinct Diurnal Cycle of Supercooled Water Cloud Fraction Dominated by Dust Extinction Coefficient, Geophys. Res. Lett., 49, e2021GL097006, https://doi.org/10.1029/2021GL097006, 2022.

Williams, E., & Renno, N.: An analysis of the conditional instability of the tropical atmosphere. Mon. Weather Rev., 121(1), 21-36, 1993.

---

## Author Comment (AC2)

**Response to Reviewer #1's Last Question:**

Yuxin Zhao et al. (Author)

**Question:** As a last question, would you have any explanation for why in Figure 2, the ISCCP cloud cover is sometimes larger than the CATS cloud cover? Do you think ISCCP is overestimating the cloud cover by e.g. mistaking aerosols for clouds? Or do you think on the contrary that CATS is somehow underestimating the cloud cover? If so, how would that be possible?

Since in the rest of the paper you consider the CATS cloud cover as the "truth", it is important to clarify this point.

**Response:** We are very appreciated for reviewer providing this important comment! We think that ISCCP sometimes overestimates the total cloud cover during daytime compared with CATS. By comparing the spatio-temporal matched total cloud cover from ISCCP, CALIPSO alone and the combined product from CALIPSO and CloudSat (that is, 2B-GEOPROF-lidar) during daytime, we find that ISCCP still overestimates the total cloud cover over TP compared with those of other space-based lidar and radar (figure not shown). Similar, Boudala and Milbrandt (2021) also found that ISCCP has larger cloud cover than that of CALIPSO over mid-latitudes (e.g., the European continent). Tzallas et al. (2019) noted that the larger cloud cover of ISCCP in the European continent is link to the relatively large viewing zenith angle (VZA) of ISCCP. Knapp et al. (2021) also suggested that there is a VZA dependence in the cloud cover of systematic errors in ISCCP results. For ISCCP, it is difficult to distinguish between aerosols and thin cirrus clouds, which may lead to spurious cloud detections and thus to an overestimation of clouds (Rossow and Schiffer, 1999). Above discussions are added in the revised manuscript (see the Line 333-343).

**Reference**

- Boudala, F. S. and Milbrandt, J. A.: Evaluations of the Climatologies of Three Latest Cloud Satellite Products Based on Passive Sensors (ISCCP-H, Two CERES) against the CALIPSO-GOCCP, Remote Sens., 13, 10.3390/rs13245150, 2021.
- Knapp, K. R., Young, A. H., Semunegus, H., Inamdar, A. K., and Hankins, W.: Adjusting ISCCP Cloud Detection to Increase Consistency of Cloud Amount and Reduce Artifacts, J. Atmos. Ocean. Technol., 38, 155-165, 10.1175/jtech-d-20-0045.1, 2021.
- Rossow, W. B. and Schiffer, R. A.: Advances in understanding clouds from ISCCP, Bull. Amer. Meteorol. Soc., 80, 2261-2287, 10.1175/1520-0477(1999)080<2261:aiucfi>2.0.co;2, 1999.
- Tzallas, V., Hatzianastassiou, N., Benas, N., Meirink, J. F., Matsoukas, C., Stackhouse, P., Jr., and Vardavas, I.: Evaluation of CLARA-A2 and ISCCP-H Cloud Cover Climate Data Records over Europe with ECA&D Ground-Based Measurements, Remote Sens., 11, 10.3390/rs11020212, 2019.